# Information content of OCO-2 oxygen A-band channels for retrieving marine liquid cloud properties

Mark Richardson[1], Graeme L. Stephens[1,2]

[1]Jet Propulsion Laboratory, California Institute of Technology, Pasadena, CA 91125, U.S.A.
[2]Department of Meteorology, University of Reading, Reading, RG6 6BB, U.K.

*Correspondence to*: Mark Richardson (markr@jpl.nasa.gov)

**Abstract.** An information content analysis is used to select channels for a marine liquid cloud retrieval using the high-spectral-resolution oxygen A-band instrument on NASA's Orbiting Carbon Observatory-2 (OCO-2). Desired retrieval properties are cloud optical depth, cloud-top pressure and cloud-pressure thickness, which is the geometric thickness expressed in hPa. Based on information content criteria we select a micro-window of 75 of the 853 functioning OCO-2 channels spanning 763.5—764.6 nm and perform a series of synthetic retrievals with perturbed initial conditions. We estimate posterior errors from the sample standard deviations and obtain ±0.75 in optical depth, ±12.9 hPa in both cloud-top pressure and cloud pressure thickness, although removing the 10 % of samples with the highest $\chi^2$ reduces posterior error in cloud-top pressure to ±2.9 hPa and cloud pressure thickness to ±2.5 hPa. The application of this retrieval to real OCO-2 measurements is briefly discussed, along with limitations and the greatest caution is urged regarding the assumption of a single homogeneous cloud layer, which is often, but not always, a reasonable approximation for marine boundary layer clouds.

## 1 Introduction

The oxygen A-band spans wavelengths with a wide range of absorption strength which can be exploited to determine photon path lengths and therefore retrieve cloud top heights and potentially the within-cloud photon path, which is related to droplet number concentration and therefore cloud thickness. Meanwhile, cloud optical depth can be retrieved from reflectance in approximately non-absorbing "continuum" channels (Fischer and Grassl, 1991a; Koelemeijer et al., 2001; Stephens and Heidinger, 2000). Such a retrieval that includes cloud geometric thickness or droplet number density would allow evaluation of model cloud physics (Bennartz, 2007). In addition A-band retrievals use reflected sunlight so are physically independent from other common sources of cloud information such as longer wavelength infrared, which may mis-identify cloud-top pressure in the presence of temperature inversions (Baum et al., 2012).

The photon path length of reflected sunlight is estimated by comparing radiance between channels with different absorption characteristics. With known absorption coefficients and similar scattering and reflection properties between the channels, the photon path length is easily determined from the Beer-Lambert Law. This technique was first suggested as a way of determining cloud-top altitude using the strong carbon dioxide ($CO_2$) absorption band near 2.0 $\mu$m with an atmospheric

window near 2.1 $\mu$m (Hanel, 1961). Subsequently the oxygen A-band near 0.76 $\mu$m was proposed as it offers improved signal to noise (SNR) and avoids overlap with the 1.87 $\mu$m water vapour absorption band (Yamamoto and Wark, 1961). It was noted that clouds are not "simple diffuse reflectors" and that "absorption along the scattering paths within the clouds must be considered".

With a single measured ratio of two channels it is only possible to determine the total photon path length and not distinguish between above-cloud and within-cloud components, as this would mean obtaining two pieces of information from a single measurement. One way of distinguishing is to take multiple measurements from diverse viewing angles, as is done by the Polarization and Directionality of the Earth's Reflectances (POLDER) instrument series (Deschamps et al., 1994). POLDER-3 has a "narrow" channel with a full width at half maximum (FWHM) of 10 nm centred at $\lambda = 763$nm, and a "wide" channel

of FWHM 40 nm centred at 765 nm. Statistics of the inferred photon path from different angles have been shown to be related to the cloud centroid pressure (Ferlay et al., 2010), results of which have been tested against CloudSat radar and Cloud-Aerosol Lidar and Infrared Pathfinder Satellite Observations (CALIPSO) data (Desmons et al., 2013). A more recent study used an information content analysis based around the characteristics of the Multiviewing, Multi-channel and Multi-polarization Imaging (3MI) and the Multiangle SpectroPolarimetric Imager (MSPI) instruments. This concluded that multiangle

measurements are informative about cloud geometric thickness, particularly for clouds thicker than 2—3 km (Merlin et al., 2016), which notably excludes the marine stratocumulus regime.

Another proposal to obtain additional measurements that inform about cloud geometric thickness is to combine measurements from both the oxygen A-band and B-band, such as those available from the Earth Polychromatic Imaging Camera (EPIC) on the Deep Space Climate Observatory (DSCOVR). By considering the sum and differences of the channel ratios it has been

proposed that cloud geometrical thickness can be retrieved when cloud optical depth ($\tau$) is greater than 5 (Yang et al., 2013).

An alternative to multiple angles or additional bands is to measure more channels in the A-band, as was done for the Scanning Imaging Absorption SpectroMeter for Atmospheric Chartography (SCIAMACHY) on board ENVISAT (Rozanov and Kokhanovsky, 2004), which when combined with the Global Ozone Monitoring Experiment instruments (GOME and GOME-2), provide an A-band record since 1995. An information content analysis based on GOME-2 characteristics, using a spectral

resolution of 0.2 nm and assumed signal-to-noise (SNR) of 100 showed that 2 pieces of information could be obtained (Schuessler et al., 2014). This study showed the best performance when retrieving cloud-top height with either $\tau$ or cloud fraction and reported that there was not sufficient information in these assumed measurements to obtain cloud geometric thickness with "satisfactory accuracy",

However, older theoretical work suggested that a spectral resolution of better than 1 cm[-1] (O'Brien and Mitchell, 1992) or even

0.5 cm[-1] (Heidinger and Stephens, 2000) is required for an effective A-band retrieval that includes cloud geometric thickness. In wavelength terms this is 0.03—0.06 nm, and is now achieved by instruments carried by the Chinese Feng-Yun 3 series (most recently FY-3D), the Japanese Greenhouse Gas Observing Satellite (GOSAT), the European Sentinel-5 Precursor (Sentinel-5P, which carries the Troposphere Measuring Instrument "TROPOMI") and NASA's Orbiting Carbon Observatory-2 (OCO-2).

This study considers OCO-2 and extends previous work that developed a lookup table to retrieve cloud-top pressure and optical depth for single layer liquid clouds over ocean (Richardson et al., 2017). This simple retrieval combined 20 of OCO-2's 853 functioning A-band channels into 2 "super-pixels" or "super-channels" based on their $O_2$ absorption. The lookup tables were used for all locations and weather conditions and were validated using collocated Moderate-resolution Imaging Spectroradiometer (MODIS) and CALIPSO data (Taylor et al., 2016). Here we develop an optimal-estimation-based retrieval (Rodgers, 2000) for single-layer water clouds over oceans using nadir-view OCO-2 measurements and subject it to several idealised tests. This study's new contributions are (i) considering information content aspects to select groups of channels rather than combined super-channels, (ii) accounting for local meteorological conditions and (iii) adding cloud pressure thickness to the retrieved state. We express cloud geometric thickness in terms of hPa and refer to it as cloud pressure thickness with the symbol $\Delta P_c$. Our current analysis considers aerosol-free cases as aerosols have not yet been properly implemented in our modified cloudy-sky version of the radiative transfer model, this is an avenue for future work and will be discussed in Sect. 5.

OCO-2 has 1,016 A-band channels of which 853 function across all soundings with spectral sampling between 0.01—0.02 nm and a FWHM of 0.04 nm in wavelength, implying sufficient spectral resolution for geometric thickness retrievals. Low marine clouds are the primary cause of spread in net modelled cloud feedback (Bony and Dufresne, 2005; Zelinka et al., 2012) and we focus on these clouds, which complements the multi-angular retrievals from other sensors which appear to perform better for thicker clouds (Ferlay et al., 2010; Merlin et al., 2016).

OCO-2 is also promising as its SNR values commonly range from 300—800 in cloudy scenes and it flies in the A-train constellation (L'Ecuyer and Jiang, 2010), allowing collocation with other sensors. Furthermore, its footprint size typically ranges from 1.2—2.3 km at nadir and compares favourably with both GOSAT (10.5 km diameter) and TROPOMI ($7 \times 7$ km$^2$), although its narrow swath of approximately 10 km is much reduced compared with TROPOMI's 2600 km.

Here we aim to develop a computationally efficient cloud retrieval for OCO-2 by selecting channels that contain the most information about the retrieved state properties, which speeds both the radiative transfer simulation and the optimal estimation calculations. In principle, the optimal channels may depend on the cloud case and on the across-track position of the measurement because the instrument line shapes (ILS) vary across the swath. Furthermore, neighbouring ILS overlap so it is more computationally efficient to select neighbouring channels since the radiative transfer will already have been calculated for many of the relevant frequencies. We refer to the selection of neighbouring channels as a "micro-window" approach and use the OCO-2 Level 2 Full Physics Radiative Transfer Model (L2FP RTM, (Boesch et al., 2015)) with a set of representative atmosphere and liquid cloud states to select the optimal micro window based on information content and posterior error criteria. This approach aims to optimise a cloud property retrieval and due to limitations related to the radiative transfer implementation and computational burden, droplet size is not a retrieved property but contributes to the posterior uncertainty. Above-cloud $CO_2$ retrievals have been found to require cloud droplet size for good accuracy (Vidot et al., 2009) and therefore our current implementation will not directly lead to above-cloud $CO_2$ retrievals.

The paper is organised as follows: Sect. 2 describes the OCO-2 satellite measurements, radiative transfer model and general information content approach. Sect. 3 details the methodology specific to this paper, including the sample atmospheres, perturbations for determining covariance matrix components, the sequential channel selection procedure and information content and retrieval analysis. Sect. 4 reports the results of each of these cases, Sect. 5 discusses the results and describes how they will be applied in the real OCO-2 cloud retrieval, and Sect. 6 concludes.

## 2 Data sources and analysis techniques

The OCO-2 satellite orbits in a Sun-synchronous orbit as part of the A-train constellation (L'Ecuyer and Jiang, 2010). It follows a 16-day repeat cycle with an equator crossing time near 13:30 in the ascending node and follows the CloudSat and CALIPSO reference ground track. OCO-2 has three viewing modes: a target mode for in-flight validation plus glint and nadir modes for operational measurements. Currently the satellite alternates nadir and glint orbits with some ocean orbits dedicated entirely to glint mode. Here we use nadir soundings to allow future cross-comparisons with the nadir-view instruments on CloudSat and CALIPSO.  Several nadir orbits pass over marine stratocumulus regions where OCO-2 offers unique value in terms of determining cloud geometric thickness for clouds that are thick enough to attenuate the CALIPSO lidar (Vaughan et al., 2009), and low enough that CloudSat suffers significantly from surface clutter (Huang et al., 2012). CloudSat measurements are further limited in terms of vertical resolution by the radar bin size which is downsampled to 240 m (Stephens et al., 2008). Currently, the main OCO-2 products are for column atmospheric CO2 concentration (XCO2 (Crisp, 2008; Crisp et al., 2016; Eldering et al., 2016; Osterman et al., 2016)) and solar-induced fluorescence (SIF, (Frankenberg et al., 2014)) which only use clear-sky soundings. Since any footprint that is identified as possibly cloudy is not processed in the standard OCO-2 products this work generates value from largely unused soundings.

OCO-2 functions in a pushbroom fashion with the footprint size depending on the viewing mode, but typically being 1.2—2.3 km. There are 8 across-track soundings, and each set of these is referred to as a frame in OCO-2 nomenclature. Within each sounding, measurements of reflected sunlight are taken in the oxygen A-band, weak-$CO_2$ and strong-$CO_2$ bands. The $CO_2$ bands are not considered in this analysis but do inform about cloud phase and droplet or particle size (Nakajima and King, 1990), and this information will be used when this retrieval is applied in our observation-based study to identify likely liquid cloud cases.

The OCO-2 A-band instrument is a bore-sighted, imaging, grating spectrometer that measures 1,016 channels spanning the wavelengths 759.2—771.8 nm. It is a flight spare from the original OCO mission and a number of focal plane array (FPA) elements have failed. 853 of the 1,016 channels are available across all soundings and over 94 % of the damaged channels occur in the A-band continuum where there is redundancy, meaning little loss of information (Richardson et al., 2017). This redundancy extends to the remaining undamaged FPA elements, meaning that fewer channels may be used to reduce the computational burden of a retrieval. The minimum number of channels required is equal to the number of elements in the retrieval state vector, provided that the channel responses to changes in the state vector properties contain orthogonal

components. Therefore, for our desired retrievals of optical depth, physical thickness and cloud-top pressure, a single cloud retrieval requires at least 3 channels. The purpose of this study is to determine how many channels are required to cover a range of realistic cloud cases and to identify those channels.

A quirk of the OCO-2 instrument complicates this determination. The wavelength of channels varies slightly between across-track soundings, which means that the sampled oxygen absorption coefficient also varies. For this reason we separately analyse each of the 8 frame sounding positions but will select a consistent micro-window of the same channels for each.

## 2.1 OCO-2 radiative transfer calculations

We use the OCO-2 Level 2 Full Physics Radiative Transfer Model (L2RTM) that was developed for the OCO-2 XCO2 retrieval. Associated wrapper code handles inputs such as interpolated ECMWF meteorological fields and accounts for the OCO-2 satellite orbit, viewing geometry and instrumental response as described in the OCO-2 data version 6 documentation (Boesch et al., 2015). The radiative transfer is based on the VLIDORT radiative transfer model with a correction for the first two orders of scattering (Natraj and Spurr, 2007; Spurr, 2006; Spurr et al., 2001) that fundamentally follows the eigenvector approach to solving the radiative transfer equation (Flatau and Stephens, 1988). This model accounts for Earth's curvature for calculating atmospheric path length of the incident and reflected solar beam, but is otherwise horizontally homogeneous. More details are provided in (Spurr, 2006) and (O'Dell, 2010).

Although the L2RTM was designed for clear-sky XCO2 retrievals, it has been validated in cloudy atmospheres by comparing OCO-2 observations with L2RTM output assuming collocated MODIS and CALIPSO cloud properties (Richardson et al., 2017). For homogeneous single-layer liquid clouds over ocean, the root mean square error (RMSE) in continuum channels was ±18 %, an overestimate of the model-only error as this includes 3d cloud effects, collocation error, parallax effects and uncertainty in the MODIS and CALIPSO retrievals.

Clouds are implemented as follows: the atmosphere is defined on 20 levels, of which one is defined as the cloud centre, one as the cloud top and one as the cloud bottom. The cloud top is placed at the cloud-top pressure and the other cloud levels are equidistantly spaced to cover the cloud-pressure thickness. An extinction coefficient is assigned to the centre level to result in the desired optical depth. Above the cloud the pressure levels are linearly interpolated from the cloud top to 1 Pa. Below the cloud they are linearly interpolated from the cloud bottom to the surface pressure. The level selected for the cloud centre is that whose pressure is closest to the cloud centre when linearly interpolated across the 20 levels from the surface pressure to 1 Pa. The L2RTM assigns extinction coefficients to layers by interpolating between levels, so a vertically homogeneous cloud layer is assumed.

Mie scattering computations are used within louds using relevant coefficients that are pre-calculated for gamma distributions of cloud droplets based on a summary of low-cloud studies (Miles et al., 2000). These values have only been pre-computed for integer values of effective droplet size. This should not affect our results greatly since our calculated uncertainties include a term spanning a range of droplet sizes. Water surfaces at nadir are dark, and even in cloud-free cases there is rarely sufficient

SNR for the OCO-2 algorithm to attempt an XCO2 retrieval. We assume a Cox-Munk surface reflectance function with the L2RTM surface reflectance set to 0.10, but as we only use nadir view over ocean there is little sensitivity to surface properties.

## 2.2 Optimal estimation and information content

We follow the principles of optimal estimation from (Rodgers, 2000), where a Bayesian retrieval combines an observation vector $\mathbf{y}$ with a prior state vector $\mathbf{x}_a$ and obtains a posterior state $\hat{\mathbf{x}}$. In our case the state vector consists of cloud-top pressure $P_{top}$, cloud pressure thickness $\Delta P_c$ and cloud optical depth $\tau$. This assumes that the observation can be related to the state by a linear forward model with some error $\boldsymbol{\epsilon}$:

$$\mathbf{y} = \mathbf{Kx} + \boldsymbol{\epsilon} \tag{1}$$

Where we refer to $\mathbf{K}$ as the Jacobian matrix as its elements are $K_{i,j} = \partial y_i / \partial x_j$. Assuming Gaussian distributions associated with $\mathbf{x}_a$ and $\mathbf{y}$, (Rodgers, 2000) shows that the best estimate of the posterior state is:

$$\hat{\mathbf{x}} = \mathbf{x}_a + \mathbf{S}_a \mathbf{K}^T (\mathbf{K} \mathbf{S}_a \mathbf{K}^T + \mathbf{S}_\epsilon)^{-1} (\mathbf{y} - \mathbf{K} \mathbf{x}_a) \tag{2}$$

And its covariance matrix is:

$$\hat{\mathbf{S}} = \left( \mathbf{K}^T \mathbf{S}_\epsilon^{-1} \mathbf{K} + \mathbf{S}_a^{-1} \right)^{-1} \tag{3}$$

Here $\mathbf{S}_a$ is the prior covariance and $\mathbf{S}_\epsilon$ the observation covariance. From Eq. (2) the posterior state $\hat{\mathbf{x}}$ is the prior $\mathbf{x}_a$ plus an iteration that is based on the difference between the observed and expected $\mathbf{y}$ with appropriate weighting for uncertainties. Eq. (3) shows that the posterior uncertainty $\hat{\mathbf{S}}$ is reduced by an amount that depends on the size of the Jacobian K weighted by the observation uncertainty $\mathbf{S}_\epsilon$. Potential nonlinearity in $\mathbf{y}(\mathbf{x})$ is addressed by iteration, with the linear expansion being determined about each iteration step.

In our OCO-2 cloud retrieval the state vector contains optical depth, cloud pressure thickness and cloud-top pressure while the observation vector is any subset of the 853 valid OCO-2 A-band channels. Using fewer channels reduces the computational burden, both in terms of the radiative transfer and for iterating the retrieval which would otherwise involve repeated inversion of 853×853 matrices.

It is common practice to select channels based on information content and/or degrees of freedom for signal (Chang et al., 2017; Mahfouf et al., 2015; Martinet et al., 2014; Rabier et al., 2002), and this approach has already been used in an oxygen A-band and B-band analysis for aerosol retrievals (Ding et al., 2016).

The information content is based on the concept of Shannon entropy and is related to the volume of state space occupied by the probability distribution $P$ that represents our knowledge:

$$S(P) = -\sum_i P(x_i) \log_2 P(x_i) \tag{4}$$

It is expressed in bits, which represents the number of binary digits required to represent the possible outcomes. A retrieval decreases the probability distribution volume, and this change in associated Shannon entropy (Shannon and Weaver, 1949) is the information content, *IC*, of the measurements:

$$IC = S(P_0) - S(P_1) \tag{5}$$

In this case $S(P_0)$ is the Shannon entropy associated with the original probability distribution and $S(P_0)$ the same value associated with the retrieved probability distribution. For multivariate Gaussian descriptions of the probability distributions, (Rodgers, 2000) shows that the information content of measurements is:

$\quad IC = \frac{1}{2}\ln|\mathbf{S_a}| - \frac{1}{2}\ln|\mathbf{\hat{S}}| = \frac{1}{2}\ln|\mathbf{S_a}\mathbf{\hat{S}^{-1}}| \tag{6}$

A related property is the degrees of freedom for signal $d_s$, which represents the number of useful independent quantities in a measurement. It may be thought of as how many different variables can be obtained from a measurement and with our three-component state vector we require a value approaching three. It may be calculated from the prior and posterior state covariances as:

$\quad d_s = tr(\mathbf{1} + \mathbf{\hat{S}}\mathbf{S_a^{-1}}) \tag{7}$

Note the different order and inversion state of the covariance matrices relative to Eq. (6). In our analysis we calculate $IC$, $d_s$ and posterior errors for continuous micro-windows of varying size and these calculations require $\mathbf{S_\epsilon}$ and $\mathbf{S_a}$. We assume prior covariances based partially on a MODIS and CALIPSO cross-validation (Richardson et al., 2017), and calculate the observation covariance $\mathbf{S_\epsilon}$ by perturbing atmospheric profiles. The calculation of the covariances is described in Sect. 3.1 and

the channel selection approach in Sect. 3.2.

While theoretically 3 channels is sufficient to retrieve 3 state vector elements, it is not clear that the same 3 channels will apply in all cases. For example, while changes in cloud-top pressure of higher clouds may lead to strong responses in channels near line cores, light in these channels may be mostly absorbed by the time it reaches lower clouds, so less-strongly absorbing channels will be preferred for lower clouds. Changes in absorption due to temperature or water vapour may also affect the

relative response of radiances to cloud properties. For this purpose, we consider a variety of atmospheric and cloud properties. Necessary observation covariances are derived by perturbing atmospheric profiles and the $IC$, $d_s$ and posterior covariance are used to select an optimal micro-window. Finally a retrieval is developed and tested on cloudy atmospheres where the "truth" is assigned and pseudo-observations and prior values are provided by sampling from the previously defined covariance matrices.

**3 Methodology, atmospheric states and cloud cases**

For ease of presentation we restrict our analysis to three representative atmospheric states, three cloud heights (680 hPa, 750 hPa and 850 hPa) and three cloud optical depths (5, 10 and 25). Together, this results in 27 combination cases. Effective droplet radius is assumed to be 12 $\mu$m, and cloud-pressure thickness is determined from the cloud geometric thickness from a subadiabatic stratiform cloud model (Borg and Bennartz, 2007):

$\quad H = \sqrt{\frac{2LWP}{c_w}} \tag{8}$

Where $C_w$ is the moist adiabatic condensate coefficient and for marine stratocumulus we use $1.9{\times}10^{-3}$ g m$^{-4}$ (range given as $1$—$2.5{\times}10^{-3}$ g m$^{-4}$ from (Brenguier, 1991)) and LWP is the liquid water path which is related to optical depth $\tau$ and effective droplet radius $r_{eff}$:

$$LWP = \frac{\tau r_{eff} 10 \rho_w}{9 Q_{ext}} \qquad (9)$$

Where $\rho_w$ is the density of water and $Q_{ext}$ the area-weighted mean scattering efficiency (Szczodrak et al., 2001), which we take to be 2. This value is chosen as it represents the large-particle limit for non-absorbing spheres (Herman, 1962) which is a reasonable approximation for cloud droplets in the oxygen A-band. Cloud geometric thickness is converted to pressure thickness by assuming that pressure decreases exponentially with altitude with a scale height of 8 km. Note that the combined Eq. (8) and Eq. (9) result comes from an adiabatic cloud model in which the LWP increases linearly with height, and differs

by a factor of 5/6 from the classic result derived for a homogeneous cloud profile (Stephens, 1978). Neither assumption is perfectly representative of reality, but the adiabatic profile is expected to be more realistic so is used here.

For the representative atmospheric states, we select all collocated soundings that are identified as single-layer liquid clouds by both MODIS and CALIPSO during November 2015 and bin them according to absolute latitude, in the ranges 0—20°, 20—50° and 50—90°. The MODIS data are from product MYD06 at 1 km horizontal resolution (Platnick et al., 2015) and the

CALIPSO data are from the 1 km resolution cloud layer product 01kmCLay (Vaughan et al., 2009). Within each bin the collocated OCO-2 ECMWF-AUX meteorological profiles (including pressure, specific humidity, temperature and wind speed) averaged level-by-level. This includes all meteorological inputs used by the L2RTM, such as pressure, temperature, humidity and wind speed.

### 3.1 Calculation of observation covariances

For simplicity we assume that the components of $\mathbf{S}_\epsilon$ are independent and consider error contributions from instrumental uncertainty $\mathbf{S}_I$, and that introduced by uncertainty in the temperature profile $\mathbf{S}_T$, humidity profile $\mathbf{S}_q$ and effective droplet radius $\boldsymbol{S_{reff}}$ such that:

$$\mathbf{S}_\epsilon = \mathbf{S}_I + \mathbf{S}_T + \mathbf{S}_q + \mathbf{S}_{reff} \qquad (10)$$

In reality, the temperature and humidity uncertainties are likely to be correlated, but this simplifies the calculation and allows

unique attribution of covariance sources. The matrix $\mathbf{S}_I$ is a diagonal matrix so averaging over more channels reduces the total posterior uncertainty even if the Jacobians are not independent. Its elements are equal to the square of the instrumental uncertainty, which depends on the radiance.

For $\mathbf{S}_T$ and $\mathbf{S}_q$ we follow the approach of (Chang et al., 2017) and perturb the tropical, mid-latitude and high-latitude atmospheric profiles 2,000 times for temperature or humidity separately with uncertainties based on 1 km resolution AIRS

validation results (Divakarla et al., 2006). For temperature we add a uniform perturbation to each level with a value sampled from a zero mean ($\mu$) Gaussian with standard deviation ($\sigma$) of $\pm1.5$ K. For specific humidity we sample from a zero mean Gaussian with a standard deviation of unity, then scale this value based on pressure level. The scaling is equivalent to $\pm20$ %

of the initial specific humidity at the surface, increasing linearly to ±50 % of the layer values at 250 hPa and remaining at ±50 % for levels with lower pressure. The calculation was also performed with 2,000 perturbations applied to $r_{eff}$ by sampling from a lognormal distribution that approximates the effective radius distribution reported by MODIS for our November 2015 low cloud cases. This lognormal fit has an arithmetic mean of 12.0 $\mu m$, but after excluding values outside the 4—30 $\mu m$

retrieved by MODIS, the arithmetic mean is 12.6 $\mu m$ and 5—95 % of the values fall within 7.5—19.4 μm. We choose $r_{eff}$ = 12 $\mu$m in our default retrieval as we are restricted to integer values by the available L2RTM Mie scattering tables, and based on its similarity to the full distribution mean.

For each set of perturbations we simulated the A-band spectra for cloud optical depths of 5, 10 and 25 and solar zenith angles of approximately 30°, 45° and 60° with a cloud-top pressure of 850 hPa. We calculate covariances at a single value of $P_{top}$,

but the convergence of our synthetic retrieval tests across a range of true $P_{top}$ values shows that we obtain reliable results regardless.

The output spectra are provided for each of the 8 different instrument line shapes associated with the 8 different OCO-2 across-track sounding positions.

For each set of 2,000 perturbed outputs, we estimated the covariance matrix elements, $S_{i,j}$, where i, j refer to channel indices,

as:

$$S_{i,j} = \sum_k (I_{i,k} - <I_i>)(I_{j,k} - <I_j>)/N \tag{11}$$

Where the sum is over the *N*=2,000 spectra of radiance *I*, which are individually referred to using the index *k*. In this case $<I_i>$ and $<I_j>$ are the sample mean radiances in the relevant channels *i* and *j*.

## 3.2 Channel selection

Eq. (3) and Eq. (6) state that we can determine the information content and posterior error covariance from the prior covariance, observation covariance and Jacobians. Our aim is to select the optimal micro-window of consecutive OCO-2 channels to provide a retrieval that efficiently reduces the posterior state error.

We use the L2FP radiative transfer model to simulate OCO-2 spectra for marine liquid clouds of $\tau$ in [5, 10, 25] and $\underline{P_{top}}$ in [680, 750, 850] hPa, for each of the 3 meteorological cases described in Sect. 3.1 and for each of the eight across-track sounding

positions. In each case, the solar zenith angle is 45° and the Jacobians for $\tau, P_{top}$ and $\Delta P$ are determined by finite differencing. The relevant observation covariance is that determined for the same sounding position, region and optical depth in Sect. 2.2 at SZA = 45°. Prior covariance is assumed to be diagonal, equivalent to an error of 1.5 in $\tau$, of 60 hPa in $P_{top}$ and of 7.5 hPa in $\Delta P$. Our $\tau$ prior error comes from applying the ±18 % error in simulated radiance for homogeneous clouds when provided with MODIS optical depth (Richardson et al., 2017). Our $P_{top}$ uncertainty is from the standard deviation of the differences between

OCO-2 and CALIPSO $P_{top}$ when using a simple lookup table for OCO-2, which we intend to use for the OCO-2 prior. The $\Delta P$ uncertainty is similar to the ±20 % error associated with Eq. (8) for clouds of cloud fraction > 0.8 reported in (Bennartz, 2007).

We consider the information content *IC*, and the 3 diagonal elements of the posterior covariance matrix $\mathbf{S}_x$. The information content accounts for non-diagonal terms in the posterior covariance, allowing an objective best selection, while the diagonal elements allow more intuitive interpretation of the magnitude of the posterior uncertainty. We refer to these using the symbol $\sigma$ with a relevant subscript, such that $\sigma_\tau^2 = S_{\tau,\tau}$ where $S_{\tau,\tau}$ is the element of the covariance matrix corresponding to the $\tau - \tau$ covariance. Note that we present the square-root of this value, i.e. $\sigma$.

This approach represents a sample of 27 unique cloud-meteorology cases across the 8 different sets of OCO-2 instrument line shapes, resulting in 216 total cases. When selecting the optimal micro-window for retrievals, it is necessary to select not just its location, but also its size (i.e. number of neighbouring channels within the micro-window).

To make this problem tractable, we select micro-windows of the following size: 5, 10, 25, 50, 75, 100, 150, 200 and 500 neighbouring channels. For each of these possible sizes we calculate *IC*, $d_s$ and the diagonal posterior error terms for every overlapping micro-window of that size. For example, the 853 individual OCO-2 channels allow 849 overlapping 5-channel micro-windows, for which we determine the information content values for each of the 216 cases.

For each size of micro-window we choose the one with the highest mean information content across the 216 cases. While this may result in a different location for each size of micro-window, the location is fixed for an individual case, i.e. the 5-channel microwindow consists of the same 5 channels in all 216 cases. We select the optimal micro-window size as that with >80 % of the 500-channel *IC*, optical depth posterior $\sigma_{\tau,\tau}$ better than ±0.05 and a posterior of better than ±1 hPa in the pressure terms $\sigma_{P_{top},P_{top}}$ and $\sigma_{\Delta P,\Delta P}$ for all 216 cases. These thresholds are by nature subjective and arbitrary.

## 3.3 Theoretical retrieval test case

We perform synthetic retrievals with known true cloud cases in mid-latitude meteorology and a 45° solar zenith angle. For each cloud case we perform 50 retrievals using a 12 micron droplet size and the prior cloud state is sampled from Gaussian distributions with $\sigma_\tau$ of ±30 %, $\sigma_{P_{top}}$ of ±60 hPa. Cloud pressure thickness is calculated from Eq. (8) with LWP from Eq. (9), and in the optimal estimation a prior $\sigma_{\Delta P}$ of ±25 % is assumed. The atmospheric humidity and temperature profiles are perturbed by sampling from the same distributions used to derive the covariance matrices in Sect. 3.2 and the observed spectrum in each case is generated by taking the simulated spectrum from the "truth" case and perturbing it by sampling from the relevant covariance matrix that has been scaled for the cloud properties according to Sect. 3.2. The squared OCO-2 radiance uncertainties are added to the diagonal elements of the observation error covariance matrix with no cross correlation. We use the standard OCO-2 version 7 uncertainties, and SNR increases as the radiance in a given channel increases. The median SNR for an individual spectrum ranges from just over 400 for the $\tau = 5$ cases to around 700 for the $\tau = 25$ cases. The single-channel SNR reaches a minimum of 72 in an absorption band channel in a $\tau = 5$ case, and a maximum of 763 in a weakly absorbing channel in a $\tau = 25$ case.

Forty true cloud cases are used with five of each case where optical depth ranges from 5 to 40 in increments of 5 and cloud-top pressure is randomly selected to be between 680—900 hPa and rounded to the nearest 10 hPa. The prior cloud properties

are assumed to be unbiased, so are randomly sampled from a Gaussian with a mean equal to the truth and a standard deviation equal to the prior errors above. Each synthetic retrieval begins with a separate prior, and the prior is also used as the first guess. The retrieval attempts assume $r_{eff}$ = 12 μm but the true $r_{eff}$ is allowed to vary and is randomly sampled from a literature summary of marine stratocumulus results, scaled to ensure a mean value of 12 μm (Miles et al., 2000). The $r_{eff}$ distribution effective

variance is fixed in each case in order to use the pre-calculated scattering properties used with the L2RTM code, but given the wide range of effective mean values considered, it is not expected that allowing the effective variance to change would greatly affect the results.

For each of the 50 perturbed prior states and observation spectra, we perform a standard 10-iteration optimal estimation retrieval (Rodgers, 2000) using the Gauss-Newton solution to optimise each step. These retrievals are done using the 75

channel micro-window selected following Sect. 3.3. The sample means and standard deviations are then compared with the known true state and indicate the theoretical performance of the micro-window retrieval.

## 4 Results

Results are presented here for the first sounding position, which is left-most when facing northwards along track during the ascending node. Our conclusions are not affected by changing the sounding position. For illustration, we select the case of

SZA = 45°, $\tau$ = 10 and $P_{top}$ = 850 hPa then present the square root of the diagonal components of covariance matrices for temperature, humidity and effective radius in Figure 1. This shows both the absolute and fractional uncertainty in the radiance due to each factor. Droplet size dominates, consistently contributing near 3 % of the radiance, although the temperature uncertainty contributes up to 1.5 % in the darker absorption channels.

Figure 2 shows the full covariance matrices for each component using the same mid-latitude meteorology, cloud properties

and SZA as Figure 1. The strongest and most consistent positive cross-correlations occur for the effective droplet size.

While the overall patterns are similar for different cloud optical depths, solar zenith angles or regional meteorology, the absolute values of the covariance matrices change. A retrieval requires an estimate of the error covariance that is relevant for the given measurement but these matrices are computationally intensive to prepare, and storing and accessing a large number of them would make the retrieval less efficient. We will therefore use a single set of retrieval matrices, one for each across-

track sounding position, and then scale the matrix to account for changes in solar zenith angle, meteorology and optical depth. Figure 3 shows the relationship between the observation covariance matrix excluding the instrumental term $\mathbf{S}_I$ for $\tau$ = 10, SZA = 30° and $\tau$ = 25, SZA = 60° with mid-latitude meteorology. Only the upper-diagonal elements of each matrix have been plotted to avoid duplication and values are scaled by $\mu_0^{-2}$, where $\mu_0 = \cos\theta_{SZA}$. There is a linear relationship between the two matrices meaning that one may be reconstructed from the other. The results are similar for tropical and high-latitude cases,

and for all soundings.

## 4.2 Micro-window selection

Figure 4 shows $IC$ and $d_s$ spectra using micro-windows consisting of 5, 75 or 200 OCO-2 channels. Also shown are the posterior errors in cloud properties taken from the square roots of the diagonal components of $\mathbf{S}_x$.

In this cloud case (mid latitude, $\tau = 10$, $P_{top} = 850$ hPa), the greatest information content comes from selecting channels near absorption features and avoiding the far wings of the A-band where only optical depth is reliably retrieved, as these channels have little $O_2$ absorption and so are uninformative about photon path length. Otherwise, the 5-channel micro-window is most sensitive to its placement within the spectrum: information content varies from 4.4—9.4 bits depending on the micro-window's location.

Micro-windows that contain fewer channels are more sensitive to changes in the instrument line shapes and cloud conditions. For example, for the 5-channel micro-window in Figure 4, the best-performing channel has an information content of 9.4 bits. However, for a different cloudy case: $\tau = 25$, $P_{top} = 680$ hPa, and for sounding position 8 instead of 1, the information content is reduced to 6.0 bits. This is a substantial loss relative to the best possible micro-window for that cloud case, which has 8.4 bits of information.

To assess the relative trade-offs between increased speed and decreased performance we take the micro-window with the highest mean information content across all cases. We then plot the central value and full range of the 216 values for each selected micro-window size in Figure 5, along with our chosen thresholds as dashed lines in each panel. The median case in the 50 channel micro-window passes our IC threshold and in all cases passes the $\tau$-uncertainty threshold, but it has multiple cases that fail the $P_{top}$ and $\Delta P_c$ thresholds. By contrast, the 75 channel micro-window containing the OCO-2 channels 353—426 (indices counting from 1 for the full 1,016 OCO-2 L1bSc channels) consistently satisfies our $P_{top}$ and $\Delta P_c$ criteria and reduces the full wavelength range from 759.2—771.8 nm to 763.5—764.6 nm.

Figure 6 shows an example cloudy scene spectrum simulated for OCO-2 and highlights the chosen 75 channel micro-window in red. Also shown is an approximated GOME-2 spectrum based on the MetOp-B instrument characteristics (Munro et al., 2016). We approximate the ILS using Gaussian instrument line shapes, taking the 0.21 nm spectral sampling from Table 1 and FWHM of 0.50 nm from Table 2 of Munro et al. (2016). While OCO-2 spectra allow 3 independent pieces of information to be obtained (see the reported $d_s$ in the figure caption) our calculations agree with previous work that the GOME-2 resolution only provides approximately 2 (Schuessler et al., 2014). Consistent with older theoretical work (Heidinger and Stephens, 2000; O'Brien and Mitchell, 1992) this analysis supports the case that OCO-2's high spectral resolution leads to additional information about cloud geometric thickness.

## 4.3 Theoretical retrieval case

Example synthetic retrieval iterations using the 75-channel micro-window are shown in Figure 7 for $\tau = 10$ and $\tau = 25$ cases, and convergence typically occurs within few iterations. Lines are coloured according to their $\chi^2$ values and it is clear that this is larger for cases where the result settles away from the true state. The posterior sample standard deviations are presented in

Table 1 for the full samples and for cases where we filter the results by excluding the 10 % of cases with the highest $\chi^2$ in each case. The greatest effect of filtering by $\chi^2$ is to reduce the uncertainty in the cloud-top pressure and cloud pressure thickness from 12.9 hPa to 2.9 hPa and 2.5 hPa respectively. The mean standard deviation in the $\tau$ retrieval is ±0.75 across all cases, but this is inflated by a large value in the $\tau = 35$ cases.

**5 Discussion**

OCO-2 $O_2$ A-band spectra are rich in information about cloud properties. Continuum channels with little absorption respond strongly to cloud optical depth, while the radiance in absorption bands is dominated by photon path length, which increases with cloud-top pressure or cloud pressure thickness. A channel's response to cloud properties depends largely on its oxygen absorption coefficient (Fischer and Grassl, 1991b; Koelemeijer et al., 2001; Stephens and Heidinger, 2000), and since many

channels have similar absorption coefficients there is redundant cloud information in OCO-2 spectra.

We ultimately selected 75 neighbouring channels as containing the majority of the cloud information. Observation covariance matrices were developed based on uncertainty related to the atmospheric temperature and humidity profiles, in cloud droplet effective radius and instrumental uncertainty. These covariances depend on the meteorological profile, solar zenith angle and cloud properties. Additionally, instrument line shapes vary across the OCO-2 swath so a separate covariance matrix is required

for each of the eight across-track OCO-2 footprints. Fortunately, when cloud or meteorological properties change, the covariance matrix elements tend to be approximately linearly related so an arbitrary covariance matrix can be reconstructed from a known case. There is greater spread in the reconstructed humidity component but this contributes a small fraction of the total covariance, which is dominated by uncertainty in the droplet radius whose component is well reconstructed.

Using 75 channels substantially reduces the retrieval processing time relative to the 853 available channels, and its usefulness

was demonstrated in a set of 8 synthetic test cases where a known cloud case was retrieved. In our perturbed tests the retrieval typically converged within 2 iterations, although a few cases converged on a local optimum instead of approaching the truth. Fortunately, these cases can generally be identified from the associated $\chi^2$, indicating that when this approach is applied to real OCO-2 data, it may be possible to flag cases where there is less confidence in the retrieval.

Our idealised posterior errors of ±0.75 in optical depth and better than ±3 hPa in cloud-top pressure and cloud pressure

thickness are based on assuming that convergence can be identified from the $\chi^2$ values, and that the cloud is single-layered and horizontally homogeneous within the OCO-2 field of view of approximately 1.4×2.2 km. This is a reasonable approximation in marine stratocumulus decks, where the typical length scale of variability in Liquid Water Path can be 10—30 km (Wood and Hartmann, 2006), but will be violated in many low level cloud cases such as at the edges of the stratocumulus-trade cumulus transition.

In addition, the assumption of a single scattering layer is commonly broken: multi-layered clouds are ubiquitous (Li et al., 2015), although for overlying cirrus it may be possible to identify and flag many of these cases based on the inferred distribution of photon path lengths from A-band measurements (Min et al., 2004). Alternatively, since OCO-2 flies in the A-train it would

also be possible to use other sensors such as CALIPSO (which is now leaving the A-train) or MODIS to identify multi-layer cloud cases, or scenes in which there is heavy aerosol loading. Cases of heavy aerosol loading are most common over the Namibian stratocumulus region with common occurrence in June-July-August (JJA) and a peak in September-October-November (SON). A combination of CALIPSO, CloudSat and International Satellite Cloud Climatology Project (ISCCP) data

imply that in the SON Namibian stratocumulus region, approximately one-third of low clouds have overlying aerosol, and approximately half of these cases are smoke (Devasthale and Thomas, 2011; Winker et al., 2010). Scattering layers overlying a marine cloud tend to reduce in the effective retrieved cloud layer pressure due to the reduced mean path length of those photons reflected from the overlying layer (Vanbauce et al., 1998). Assessment of aerosol effects will be necessary in future work.

It was also assumed that the clouds will be reliably identified as liquid, and that a constant effective droplet size may be assumed. Droplet size variance has been included in terms of the observation covariance, but this limits our retrieved posterior covariance. Cloud identification is relatively simple for nadir A-band reflectance measurements over ocean, as for most solar zenith angles the surface is dark and cloudy scenes may simply be identified when reflectance exceeds some threshold. The OCO-2 instrument also carries weak- and strong-CO2 band spectrometers, and with ice absorbing more strongly than water in

the near infrared we will be able to use well-known retrieval principles to obtain cloud phase (Nakajima and King, 1990).

Our assumptions mean that the true error of an OCO-2 based cloud retrieval will be larger than that reported here, but our results suggest that the use of a 75-channel micro-window is justified as the basis of an OCO-2 cloud retrieval for marine liquid cloud properties.

## 6 Conclusions

The OCO-2 satellite carries an $O_2$ A-band spectroradiometer with high spectral sampling. Our analysis supports that this spectral sampling is sufficient to, in principle, allow determination of the optical depth, cloud-top pressure and geometric pressure thickness of clouds. It has been demonstrated that observed OCO-2 spectra respond largely as expected to changes in cloud optical depth and cloud-top pressure (Richardson et al., 2017), but that study did not use modern Bayesian techniques. Such techniques account for relevant conditions such as line broadening due to local meteorology and they also account for

prior information and cross-correlation between the responses of individual channels.

Here we report that the OCO-2 A-band spectra contain much redundant information as a number of channels experience similar oxygen absorption. After accounting for observational errors associated with uncertainty introduced by meteorology, cloud droplet size and instrumental error, it was found that with a micro-window of 75 continuous channels, most of the information from the full 853-channel spectrum is retained. In a perfectly linear theoretical case, posterior error in cloud-top pressure and

cloud pressure thickness were reduced below ±1 hPa and optical depth below ±0.05.

Using perturbed synthetic tests, the majority of cases approached the known truth and the full sample posterior errors averaged ±0.75 in optical depth, ±12.9 hPa in $P_{top}$ and cloud pressure thickness. Cases that converged to a state away from the truth

could generally be identified by their large $\chi^2$ values, and removing the 10 % of worst cases reduced the posterior sample standard deviation in $P_{top}$ and $\Delta P_c$ was reduced to $\pm2.9$ and $\pm2.5$ hPa.

These results apply in an ideal theoretical case of a uniform single layer liquid droplet cloud, and retrieval errors will be larger in reality where these assumptions do not apply. However, violations of these assumptions such as real-world cloud heterogeneity, will likely have a similar effect on both the full spectrum and on our selected 75-channel micro-window. We therefore propose that these assumptions do not affect our primary conclusion regarding the relative performance of our optimised retrieval versus a more intensive, full spectrum retrieval.

**Acknowledgments** This research described in this paper was performed at the Jet Propulsion Laboratory, California Institute of Technology sponsored by NASA. MR has been funded by the OCO-2 and CloudSat projects. MR would like to thank James McDuffie and Jussi Leinonen for providing radiative transfer and optimal estimation code assistance, plus Matt Lebsock, Annmarie Eldering, Mike Gunson, Chris O'Dell, Tommy Taylor, Heather Cronk and Aronne Merrelli for helpful technical discussions. The OCO-2 science data are available online from the NASA Goddard GES DISC at https://disc.gsfc.nasa.gov/datacollection/OCO2_L1B_Science_7.html.

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

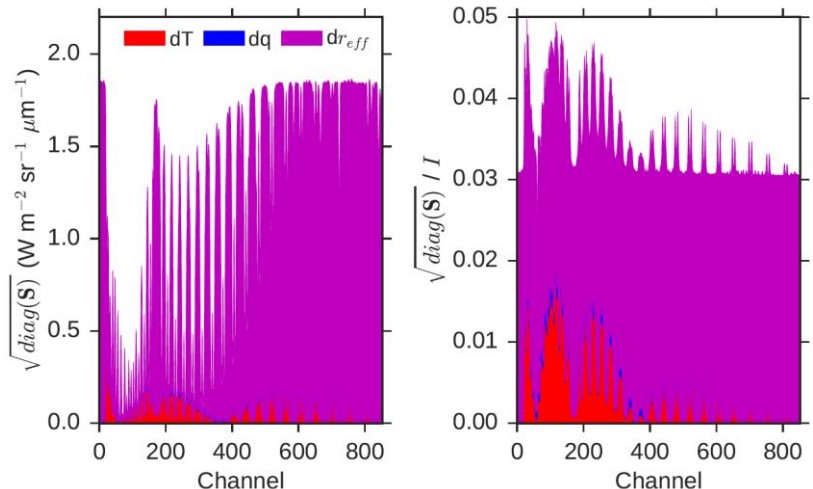

**Figure 1 Square-root of diagonal components of the covariance matrix, stacked contribution from temperature (red), humidity (blue) and effective radius (magenta). Results shown for a cloud with $\tau = 10$ and $P_{top} = 850$ hPa. Left shows the value in absolute radiance, and right as a fraction of the unperturbed radiance such that 0.03 represents an uncertainty of ±3 %.**

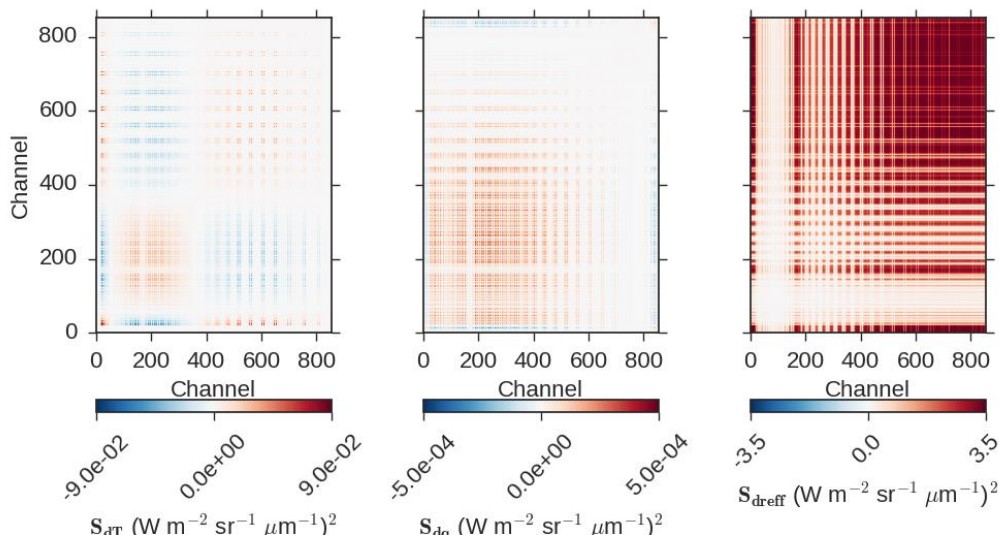

**Figure 2 Example covariance matrices for each component as labelled in the colour bar: (left) temperature, (middle) humidity, (right) effective radius.**

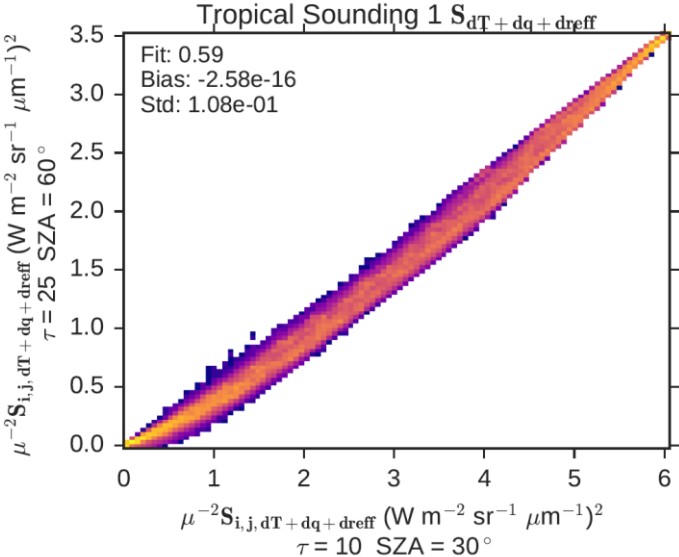

Figure 3 2d histograms of covariance matrix elements at 60° solar zenith angle as a function of the same values at 30° solar zenith angle for the mid-latitude meteorological state, each value has been scaled by $\mu_0^{-2} = cos^{-2}(SZA)$ to account for differences in illumination geometry.

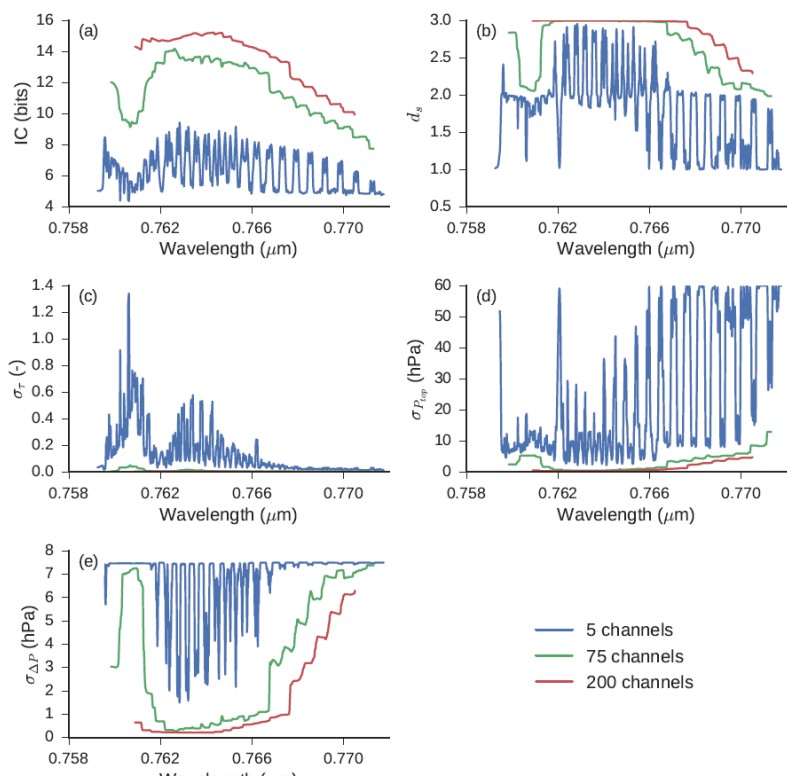

**Figure 4 Results of information content analysis for a $\tau = 10$ and $P_{top} = 850$ hPa cloud in mean mid-latitude meteorology for OCO-2 sounding position 1. Each line represents the result using a micro-window of difference size, centred on the OCO-2 channel given in the x-axis. Values are as follows: (a) information content in bits, (b) degrees of freedom for signal, (c—e) square root of diagonal elements of posterior state covariance matrix, (c) cloud optical depth, (d) cloud-top pressure and (e) cloud pressure thickness.**

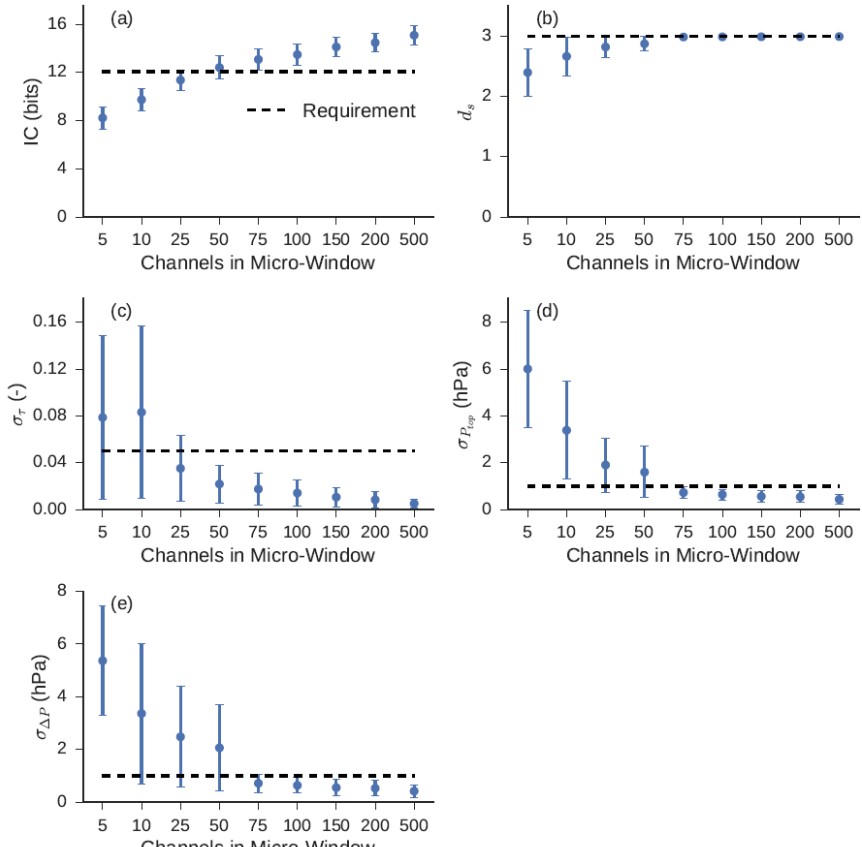

**Figure 5 Range of performance for best-located micro-window of each size. The point represents the central value and the lines the full range of the 216 outputs covering each sounding position, meteorology and cloud case. Note that the x-axis is non-linear. (a) Information content in bits, (b) degrees of freedom for signal, (c—e) square root of diagonal covariance matrix elements: (c) cloud optical depth, (d) cloud-top pressure and (e) cloud pressure thickness. Dashed lines represent selected retrieval requirements.**

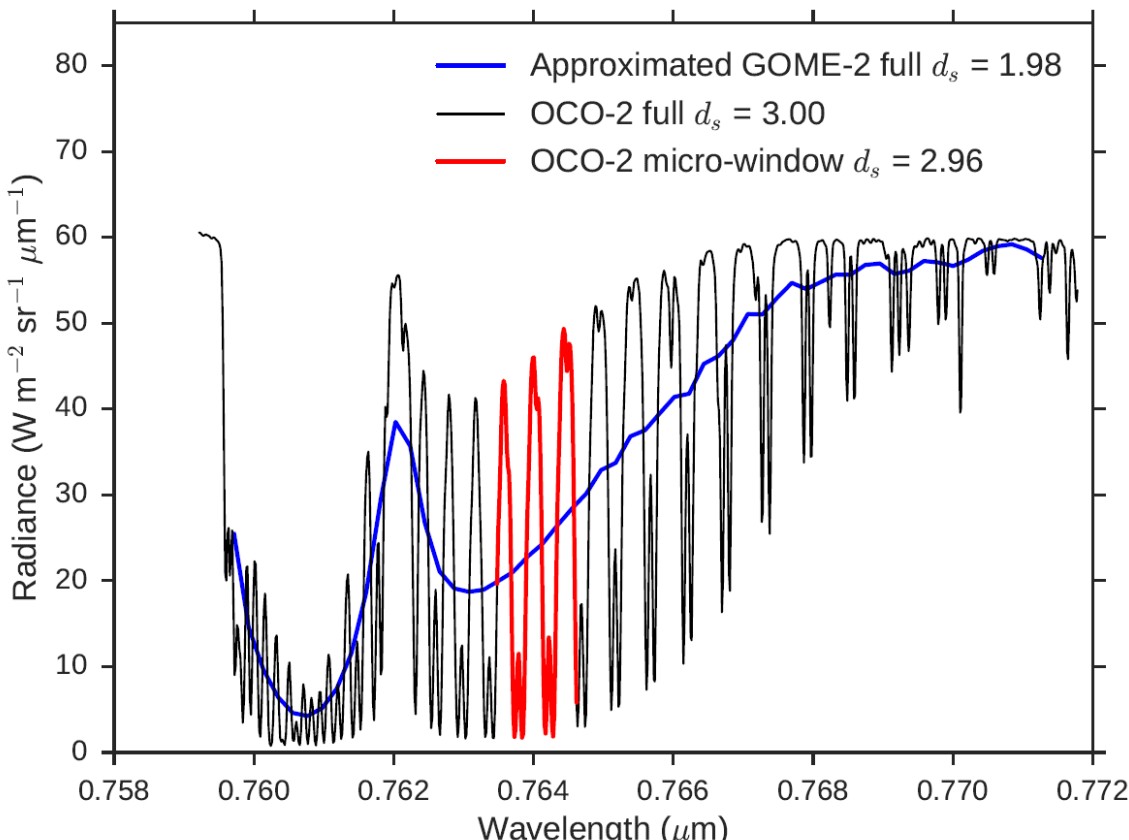

**Figure 6 Example simulated cloudy scene A-band spectrum, for a $\tau = 10$, $P_{top} = 850$ hPa cloud in a tropical atmosphere with a solar zenith angle of 45°. The black line shows the full OCO-2 simulated spectrum, the blue line is the black line resampled using approximate GOME-2 instrument line shapes and the red line is the selected 75 channel micro-window for OCO-2 cloud retrievals. The legend also reports the $d_s$ for each spectrum with the GOME-2 instrumental uncertainty based on an SNR of 100 as in previous work (Schuessler et al., 2014).**

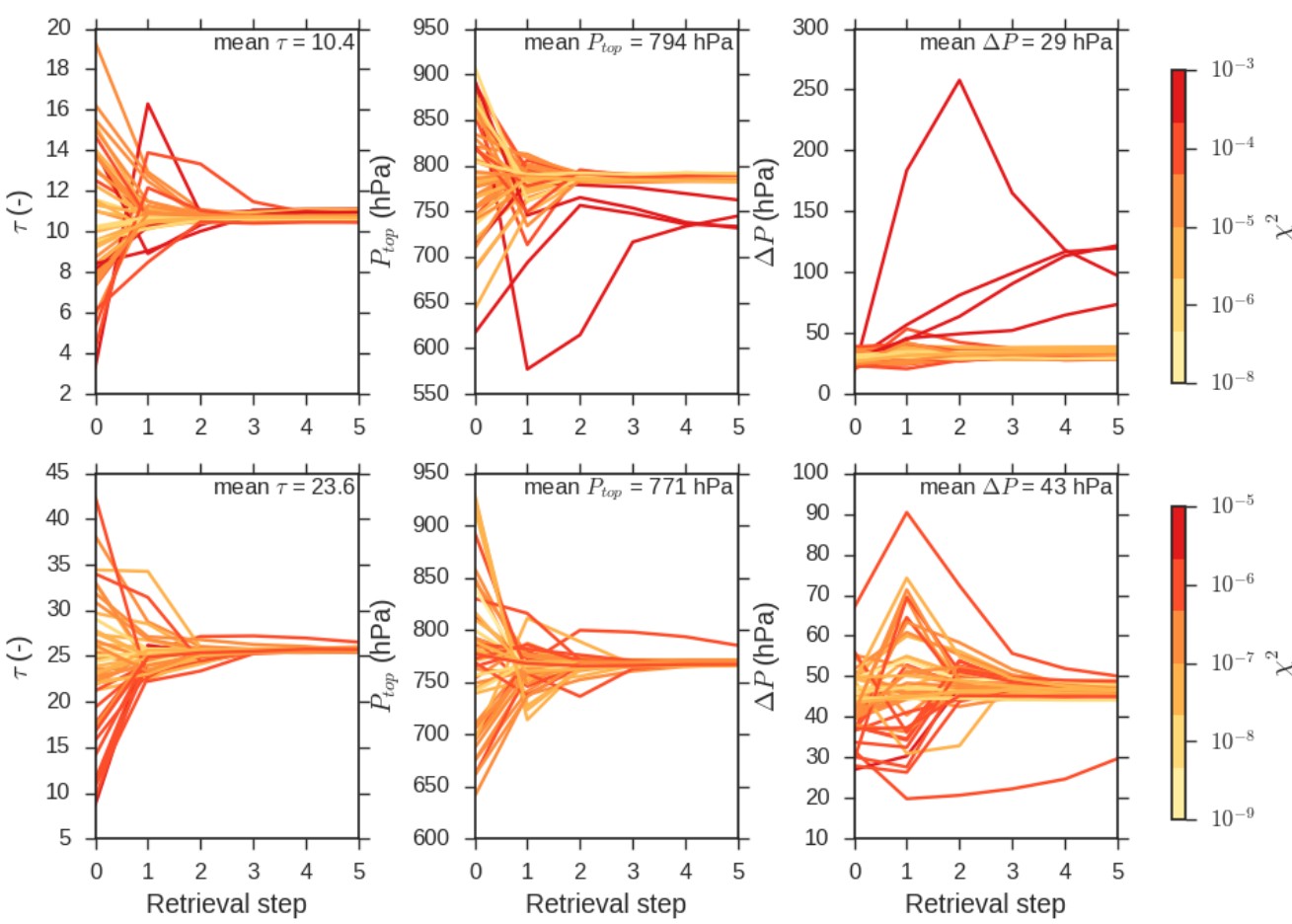

**Figure 7 Example iteration in retrieved cloud properties for synthetic micro-window retrievals for test cases with cloud optical depth near 10 (top) and 25 (bottom). Each line represents the iterations through one of the 50 sample retrievals. The lines are coloured according to their $\chi^2$ values, note the separate colour bars for the top and bottom rows with larger $\chi^2$ for the top cases.**

**Table 1 – Posterior errors estimated from the sample standard deviations of the retrieval output. In each case, $\sigma$ refers to the full sample standard deviation and $\sigma$ (filtered) refers to the sample standard deviation excluding those with the 10 % highest values of $\chi^2$. The bottom row shows the mean of the standard deviations entered in each row.**

| | $\tau$ | | ctP (hPa) | | dP (hPa) | |
|---|---|---|---|---|---|---|
| True $\tau$ | $\sigma$ | $\sigma$ (filtered) | $\sigma$ | $\sigma$ (filtered) | $\sigma$ | $\sigma$ (filtered) |
| 5 | 0.22 | 0.22 | 6.5 | 6.3 | 5.3 | 5.3 |
| 10 | 0.27 | 0.25 | 13.7 | 2.3 | 22.0 | 2.5 |
| 15 | 0.46 | 0.45 | 1.8 | 1.8 | 1.0 | 1.0 |
| 20 | 0.15 | 0.15 | 1.3 | 1.3 | 0.8 | 0.8 |
| 25 | 0.13 | 0.13 | 1.4 | 1.3 | 0.9 | 1.0 |
| 30 | 0.32 | 0.32 | 1.5 | 1.4 | 1.2 | 1.1 |
| 35 | 7.52 | 1.95 | 19.5 | 2.9 | 11.3 | 3.1 |
| 40 | 15.27 | 0.49 | 26.7 | 2.3 | 26.4 | 0.9 |
| Average | 6.02 | 0.75 | 12.9 | 2.9 | 12.9 | 2.5 |