# Peer review of "Information content of OCO-2 oxygen A-band channels for retrieving marine liquid cloud properties"

_Atmospheric Measurement Techniques, 2017_

## Referee Comment (RC1) · Anonymous Referee #2 · 18 Oct 2017

Detailed review on the paper: Information content of OCO-2 oxygen A-band channels for retrieving marine liquid cloud properties.

**I. General comments**

I think this paper is very interesting and brings innovation on how to retrieve cloud properties with OCO-2. The use of optimal estimation method makes the study very robust.

I have some remarks concerning the introduction. I think you should rework it to make it more complete. Indeed you should answer the following questions:

- What are the motivations for this study?
- What has already been done?
- What does your study bring?

As those aspects are not clear. I also find your bibliography too light. We don't expect you to quote all the works done in the O2 A-band and optimal estimation, but at least some of them. You can read the paper of Merlin et al (2017) as the subject is close to yours and the bibliography is quite complete.

**II. Specific comments**

p1
L 19-20, there are numerous papers that you can quote.

p2
l25: multiply scatter : not nice
l25-26-27-28: This sentence is too long
l31: This work ....: Sentence not clear

p3
l4: do contain information.... Reference is missing
l21 ECMWF meteorological fields : Reference missing

p4
l18 observed and expected y : is a value missing after "observed"?
l15 to 30: When you refer to a vector or a value you could write its symbol
l22 observation vector instead of observation state vector
l22 a point is missing after channels
l27 Shannon entropy : Reference missing

p5
l1: You don't define $P_0$ and $P_1$
l6 :see my comment p4 l15
l19 : Methodology **and** example atmosphere **and** cloud ..
Not nice.

p6
l1 $\rho_w$ not present in eq 8
l7: Why do you take Qext =2?
l7: 0°-20°, 20°-50° and 50°-90°, you forgot the degree symbol over 0, 20 and 50.
l7: 'identified as single-layer liquid clouds by both MODIS and CaLiPSO'. It may be useful for the reader to write which product/ collection you used.
l8-9: You should rewrite the 2 sentences which are not clear.  For instance :
'Within each bin, all the OCO-2 ECMWF-Aux profiles (including pressure, temperature, humidity and wind speed) are averaged level-by level.'

l22:  not nice. You should rewrite the description of the uncertainties, particularly for the humidity.
l25: standard deviation **of** +-1.5K
we sample: what are you sampling?
l26: with 2000 perturbations **applied**  to reff
l27: '5--95% range of 7.5--19.4 um' Not sure of what it means. Try to avoid the abbreviations in the text and write a sentence.
l29: The output was **sampled**: You are using this word quite often and maybe not always with the right sense?

p7
l8: cases **described** in sect. 3.1
l12: not nice: to an error of 1.5 on $\tau$, of 60hPa on Ptop and of 7.5hPa on $\Delta P$
l14: Our uncertainty is approximately: What does it mean?
l18-19: 'more intuitive': not very nice, more qualitative ?

p9
Description of figure 3: I am confused as the caption seems to say that there are two figures (top and bottom), but only one is visible.

Description of figure 4: I don't know where to see the channels you are mentioning (l9) as the plot is in function of the OCO-pixels. It might be a good idea to show a spectra of OCO lines.

Description of figure 5:
l20 content**2**. remove the 2.
l23: Again showing a spectra with your selected window might be a good idea.

Also How did you choose the thresholds?
You should justify more the choice of 75p as it is not obvious from the plot. 50p could be fine also?

p10
l2-3: Once again, showing a spectra would help the reader to follow your conclusions.
l9-10-11: Sentence too long.

III. Technical corrections

- When you quote a paper within a sentence (p2 l3) you shouldn't put the author's name between parentheses.
This study goes beyond Richardson et al (2017) by ....

- I don't know what is the AMT policy for that but it would be better to centre your equations.
- In the bibliography, you might think to put the first authors in bold and the titles in Italic; otherwise it is very difficult to distinguish the different papers.
- Figures: In general, be careful with the size of the axis-labels which are very small (fig 2 , 4)
- The numbers of the lines restart at 0 at each page, I don't know if it is a mistake or not.

---

## Referee Comment (RC2) · Anonymous Referee #1 · 18 Oct 2017

Review of Richardson & Stephens paper:

This is a very interesting and valuable study. I would be very interested to know how this study could transfer to airborne spectrometers like AVIRIS and PICARD that also have high spectral resolution and lack IR channels for cloud top retrieval. We've done a similar thing with ASTER: used an instrument that was previously only for clear-sky work and created a product from unused data.  The paper is overall well written and methods are clearly described and understandable.

**Major comments:**

Marine SCu frequently have some kind of aerosol sitting on top of them especially off the coast of Africa (Sahara dust and Namibia smoke) and to a lesser extent the US Pacific Coast (mostly smoke). Have you tried inserting above-cloud aerosol layers into your simulations and seeing what happens? I'm not saying that you have to correct for aerosols but some idea as to uncertainty introduced by absorbing aerosols would be nice.

Please be consistent in definition of micro-window. You use "pixels" in the first 8.5 pages of the paper and then switch to "channels" for the rest of the text. I personally would prefer you use "channels", but you can use whichever you see fit as long as it's consistent throughout.

**Minor comments:**

Figure 3 caption should read $\mu_0^{-2}=\cos^{-2}(SZA)$, $\mu$ is normally used to indicate sensor zenith angle.

Page 1 Line 1: please expand CALIPSO acronym, first use
Page 2 Line 21: should read "equator crossing time near 13:30"
Page 7 Line 25: please clarify what the micro-windows are measured in: 500 of what? Later in the text, on page 9 it becomes clear that the units of the micro-window size are channels. For folks that don't normally use something like OCO, it might help giving a bit more information, like what a 75-channel micro-window translates into as far as a wavelength range goes. It would make the research more transferable to other instruments as this is a potentially very valuable retrieval approach.
Page 9 Line 3: please use $\theta_0$ and $\mu_0$ as is generally customary for solar zenith angle and its cosine
Page 9 Line 20: "highest mean information content2. " A typo?
Page 10 Line 26: OCO is in the constellation with Aqua, so you may be able to use the MODIS multilayer cloud map in order to stay away from cirrus. That's just what that map is for.

---

## Referee Comment (RC3) · Anonymous Referee #3 · 24 Oct 2017

Review comments on manuscript "Information content of OCO-2 oxygen A-band channels for retrieving marine liquid cloud properties"

Authors: M. Richardson and G. L. Stephens

MS No.: amt-2017-314

MS Type: Research article

General comments:

This paper presents a theoretical study on retrieving marine boundary layer cloud optical thickness, pressure thickness, and top pressure, using the OCO-2 oxygen A-band

measurements. The method is well defined and the results are of interests to the community. The topic is suitable for publication in AMT, but I do have some concerns for the authors to consider.

1) Marine boundary layer clouds are targets that we have pretty good a priori knowledge; hence it's not surprising to have good retrieval accuracy, but since the goal of the research is to apply the method to OCO-2 retrievals, one question would be how to decide when to retrieve? I would suggest adding at least some discussions on how to identify the clouds that are suitable for applying this method.

2) The literature review should have been more complete. There have been studies on retrieving cloud pressure thickness plus cloud top pressure in the past, especially for thick clouds over dark surfaces (e.g., Ferlay et al. 2010, Yang et al. 2013, Merlin et al., 2016, reference given below).

3) I found the structure of the paper makes understanding the contents difficult. I would suggest some re-arrangements. For example, Section 2 is titled "The OCO-2 satellite and its instruments", I couldn't see how the two subsections fit there: " 2.1 OCO-2 radiative transfer calculations" and "2.2 Optimal estimation and information content". My suggestion would be to use one section to describe forward modeling issues and another section for retrieval related issues.

4) I would suggest converting the information content shown in the article to how many pieces of information can be retrieved. For example, it's not clear to me what information content = 16 means (the red line in Figure 4(a)) physically.

References:

Ferlay, N., and F. Thieuleux, C. Cornet, and A. B. Davis, 2010: Toward New Inferences about Cloud Structures from Multidirectional Measurements in the Oxygen A Band: Middle-of-Cloud Pressure and Cloud Geometrical Thickness from POLDER-3/PARASOL. J. Appl. Meteor. Climatol., 49, 2492–2507. doi:

http://dx.doi.org/10.1175/2010JAMC2550.1.

Merlin, G., Riedi, J., Labonnote, L. C., Cornet, C., Davis, A. B., Dubuisson, P., .Parol, F., 2016: Cloud information content analysis of multi-angular measurements in the oxygen A-band: Application to 3MI and MSPI. Atmospheric Measurement Techniques, 9(10), 4977-4995. doi:http://dx.doi.org/10.5194/amt-9-4977-2016.

Yang, Y., A. Marshak, J. Mao, A. Lyapustin, J. Herman, 2013: A Method of Retrieving Cloud Top Height and Cloud Geometrical Thickness with Oxygen A and B bands for the Deep Space Climate Observatory (DSCOVR) Mission: Radiative Transfer Simulations. J. Quant. Spectrosc. Radiat. Trans. 122, 141-149, http://dx.doi.org/10.1016/j.jqsrt.2012.09.017.

---

## Referee Comment (RC4) · Anonymous Referee #4 · 30 Oct 2017

This paper analyzed the information content in O2 A band for retrieving marine liquid cloud properties. it used the Rodgers (2000) formal optimization framework and expressed the information content in terms of degree of freedom for signal and Shannon entropy. The O2 A band on OCO-2 has 800+ channels, and this paper shows that only ~75 channels are needed to retain all information content for retrieving cloud optical depth, cloud pressure thickness, and cloud-top pressure. The method in this paper is sound, but revisions are needed to include various advances in recent studies of using O2 A and B for cloud/aerosol height retrievals, as well as more justification about assumptions and caveats in this study.

1. Abstract. what is cloud-pressure thickness? what is the unit here?

2. Introduction. Most references cited in the first paragraph are theoretic work done in the past. While they are interesting, there are renewed interests in recent years to use O2 A and B band to retrieve cloud/aerosol height, with some using real data with good validations. They should be included in this paper, and discussion should be made that recent studies with real data use only O2 A/B bands from an imager (such as EPIC or MERES), although some studies did recommend the use of spectra to retrieve aerosol height. See references below and references therein (including some work done by authors' colleagues in JPL).

Ding, S. et al., 2016, Polarimetric remote sensing in O2 A and B bands: Sensitivity study and information content analysis for vertical profile of aerosols, Atmospheric Measurement Techniques, 9, 2077-2092.

Xu, X. et al., 2017, Passive remote sensing of altitude and optical depth of dust plumes using the oxygen A and B bands: First results from EPIC/DSCOVR at Lagrange-1 point, Geophys. Res. Lett., 44, 7544-7554.

3. Section 2. It should be made clear if cloud properties are well characterized, will CO2 be retrieved accurately in cloudy-sky conditions? If so, is it column CO2 above cloud top or whole atmospheric column, including CO2 within cloud? Any references will be helpful in this regard. To what degree of accuracy of cloud properties are needed in order to retrieve CO2 with good accuracy?

4. Section 2.1. It is noted that there are often aerosol layer above marine boundary layer cloud. To be clear, no aerosol effects are treated in L2RTM, correcT? How about surface reflectance?

section 2.2. what is the state vector? optical depth, top pressure/height? be clear here. This also applies to the title of this paper. what properties to be retrieved? droplet size, top pressure/height or optical depth?

5. Page 4, Line 25. Ding et al. (2016) used similar method to select channels needed in O2 A and B band for aerosol retrievals. It is worthy to mention here.

6. Page 5, L20. How about effective variance of cloud droplet size? Does it matter?

7. Page 6, L2. "A pressure scale height of 8 km is assumed to convert the resultant ....". This sentence is hard to comprehend.

8. Page 6, L27. mean of 12.6 um? should it be 12 um to be consistent with previously stated? How about effective variance?

9. Page 6, L29. cloud top pressure of 850 hpa? but, in the 3.1, it says three different pressures.

10. Page 8, L10. what is the priori for cloud top pressure here? what is the error in OCO-2 measurement itself?

11. P10, L7. Do these 75 channels have the same wavelenghts for all test cases?

12. P11, last sentence. what is proposed here is a strong statement. What is the basis to support that "assumptions made here don't affect primary conclusion" here?

13. Finally, it is not all that clear if measurement in O2 A with such a finer spectral resolution will be needed? In other words, using 75 channels vs. using just one channel (such as from EPIC, MERES or TROPOMI) for cloud retrievals, are there huge differences? Answering this question will greatly improve the impact of this paper.
* * *

---

## Author Comment (AC1) · 8 Jan 2018

**General response to reviewer 1**
We have responded to each of your points below, with your text in red and ours in blue and believe we have addressed your major concerns. We did not act on some of your minor suggestions but have justified this in each case. Typically this is because of linguistic style choices or because of the AMTD template.

The largest changes made in response to your comments are that the introduction has been greatly extended and we have added a new Figure 6. This contains an example OCO-2 spectrum, highlights our micro-window and also shows a GOME-2-like spectrum. These make the paper much more accessible and allow much easier comparison with other instruments.

It is obvious that you read our submission with great attention, thank you for your time and feedback.

**NOTE:** our page and line numbers refer to the new version. With our greatly expanded introduction and other minor corrections it became very messy otherwise. The newly added Figure 6 is appended at the end of the text.

**Detailed review on the paper: Information content of OCO-2 oxygen**
A-band channels for retrieving marine liquid cloud properties.

I. General comments
I think this paper is very interesting and brings innovation on how to retrieve cloud properties with OCO-2. The use of optimal estimation method makes the study very robust.

I have some remarks concerning the introduction. I think you should rework it to make it more complete. Indeed you should answer the following questions:

- What are the motivations for this study?
- What has already been done?
- What does your study bring?

As those aspects are not clear. I also find your bibliography too light. We don't expect you to quote all the works done in the O2 A-band and optimal estimation, but at least some of them. You can read the paper of Merlin et al (2017) as the subject is close to yours and the bibliography is quite complete.

➔ **Response:** We tried to keep the paper concise, but now agree that we missed too much context so have made major changes.
➔ **Changes made:** Much rewritten and added text, covering p1L18—p3L34. The introduction has been rewritten and lengthened with citations to Hanel (1961), Yamamoto & Wark (1961), Deschamps et al. (1994), Ferlay et al. (2010), Desmons et al. (2013), Merlin et al. (2016), Yang et al. (2013), Rozanov & Kokhanovsky (2004), Schuessler et al. (2014), Heidinger & Stephens (2000) and O'Brien & Mitchell (1992). These support a new summary of various A-band cloud studies and then justify our new work as applying hyperspectral approaches that are useful for low clouds. We cite Bony & Dufresne (2005) and Zelinka et al. (2012) to support the importance of low clouds that are poorly sampled by the multi-angular approaches, and explain our advantages for geometrical thickness relative to other work that used instruments with lower SNR and spectral resolution.

II. Specific comments

p1
L 19-20, there are numerous papers that you can quote.

➔ **Response:** See changes above.
➔ **Changes made:** Introduction fully rewritten.

p2
l25: multiply scatter : not nice

➔ **Response:** Term deleted, the lidar being attenuated justifies the point on its own.

➔ **Changes made:** "…attenuate and multiply scatter the CALIPSO lidar" ➔ "attenuate the CALIPSO lidar"

l25-26-27-28: This sentence is too long
➔ **Response:** Agreed.
➔ **Changes made:** Sentence split into two.

l31: This work ....: Sentence not clear

➔ **Response:** Justification added.
➔ **Changes made:** Sentence now reads: "Since any footprint that is identified as possibly cloudy is not processed in the standard OCO-2 products this work generates value from largely unused soundings."

p3
l4: do contain information.... Reference is missing

➔ **Response:** This is based on Nakajima-King-like principles but I don't have the formal information content analysis for the OCO-2 instrument. Therefore we changed the wording slightly and added a citation.
➔ **Changes made:** p4L22—25 changed and now reads "The CO2 bands are not considered in this analysis but do inform about cloud phase and droplet or particle size (Nakajima and King, 1990), and this information will be used when this retrieval is applied in our observation-based study to identify likely liquid cloud cases."

l21 ECMWF meteorological fields : Reference missing

➔ **Response:**
➔ **Changes made:** p5L10—11 added text: "response as described in the OCO-2 data version 6 documentation (Boesch et al., 2015)"

p4
l18 observed and expected y : is a value missing after "observed"?

➔ **Response:** The meaning is intended as "observed y and expected y" but that feels clunky to me. Another option is to hyphenate to "observed- and expected y", but grammar guides now disagree over that use and it seems archaic. I thought context made it clear, but have added a little extra text to further emphasise the context.
➔ **Changes made:** p6L14—15 rewritten slightly to: "based on the difference between the observed and expected y"

l15 to 30: When you refer to a vector or a value you could write its symbol

➔ **Response:** Symbols added to aid the reader, with minor rephrasing so that it's clear that S-hat refers to the posterior uncertainty and not the "reduction in posterior uncertainty".
➔ **Changes made:** Vector and matrix symbols added and text changed, e.g. "reduction in posterior uncertainty" ➔ "posterior uncertainty $\hat{S}$ is reduced by…"

l22 observation vector instead of observation state vector

➔ **Response:**
➔ **Changes made:** change made.

l22 a point is missing after channels

➔ **Response:**
➔ **Changes made:** change made.

l27 Shannon entropy : Reference missing

➔ **Response:**

➔ **Changes made:** p6L30 now reads "…and this change in associated Shannon entropy (Shannon and Weaver, 1949)…"

**p5**
**l1: You don't define P0 and P1**

➔ **Response:**
➔ **Changes made:** p7L2—4 now reads "In this case $S(P_0)$ is the Shannon entropy associated with the original probability distribution and $S(P_0)$ the same value associated with the retrieved probability distribution."

**l6 :see my comment p4 l15**

➔ **Response:**
➔ **Changes made:** Symbol added.

**l19 : Methodology and example atmosphere and cloud ..**
**Not nice.**

➔ **Response:**
➔ **Changes made:** Changed to "Methodology, atmospheric states and cloud cases"

**p6**
**l1 ρw not present in eq 8**

➔ **Response:** Good catch, this was a typo.
➔ **Changes made:** \rho converted to \rho_w in Equation 8.

**l7: Why do you take Qext =2?**

➔ **Response:** Size parameters $x = 2\pi r/\lambda$ here are >50 and water is weakly absorbing (real part of index ~1.33, imaginary part ~1×10⁻⁷), so I take $lim_{x\to\infty}$ case for a non-absorbing sphere.
➔ **Changes made:** p8L6—7 text added: "This value is chosen as it represents the large-particle limit for non-absorbing spheres (Herman, 1962) which is a reasonable approximation for cloud droplets in the oxygen A-band"

**l7: 0°-20°, 20°-50° and 50°-90°, you forgot the degree symbol over 0,**
**20 and 50.**

➔ **Response:** This appears to be an AMT style choice. Under "English guidelines and house standards" it says "**En dashes (–)** are longer than hyphens (-) and serve numerous purposes….En dashes are used to indicate, among other things, relationships (e.g. ocean–atmosphere exchange), ranges (e.g. 12–20 months),…" this implies that for ranges the unit follows the latter value only.
➔ **Changes made:** None

**l7: 'identified as single-layer liquid clouds by both MODIS and**
**CaLiPSO'. It may be useful for the reader to write which product/**
**collection you used.**

➔ **Response:**
➔ **Changes made:** p8L14—15 now reads: "The MODIS data are from product MYD06 at 1 km horizontal resolution (Platnick et al., 2015) and the CALIPSO data are from the 1 km resolution cloud layer product 01kmCLay (Vaughan et al., 2009)."

**l8-9: You should rewrite the 2 sentences which are not clear. For**
**instance :**
**'Within each bin, all the OCO-2 ECMWF-Aux profiles (including pressure, temperature, humidity and wind speed) are averaged level by level.'**

➔ **Response:** Agreed.

➔ **Changes made:** p8L15—17 now use your suggested text.

l22: not nice. You should rewrite the description of the uncertainties, particularly for the humidity.

➔ **Response:** The humidity method description was split by the temperature sampling description, we've rewritten to ensure that the specific humidity perturbations are described continuously and hope that this is clearer.
➔ **Changes made:** p8L30—p9L2 now reads: "For temperature we add a uniform perturbation to each level with a value sampled from a zero mean (µ) Gaussian with standard deviation (σ) of ±1.5 K. For specific humidity we sample from a zero mean Gaussian with a standard deviation of unity, then scale this value based on pressure level. The scaling is equivalent to ±20 % of the initial specific humidity at the surface, increasing linearly to ±50 % of the layer values at 250 hPa and remaining at ±50 % for levels with lower pressure."

l25: standard deviation **of** +-1.5K
we sample: what are you sampling?

➔ **Response:** Above text change hopefully addresses this.
➔ **Changes made:** See above.

l26: with 2000 perturbations **applied** to reff

➔ **Response**.
➔ **Changes made:** "applied" added.

l27: '5--95% range of 7.5--19.4 um' Not sure of what it means. Try to avoid the abbreviations in the text and write a sentence.

➔ **Response:** We have rewritten this in a way that we hope is clearer.
➔ **Changes made:** p9L4—5 now reads: "This lognormal fit has an arithmetic mean of 12.0 µm, but after excluding values outside the 4—30 µm retrieved by MODIS, the arithmetic mean is 12.6 µm and 5—95 % of the values fall within 7.5—19.4 µm."

l29: The output was **sampled**: You are using this word quite often and maybe not always with the right sense?

➔ **Response:** Agreed.
➔ **Changes made:** p9L12—13 now reads: " The output spectra are calculated for each of the 8 different instrument line shapes associated with the 8 different OCO-2 across-track sounding positions"

p7
l8: cases **described** in sect. 3.1

➔ **Response:** Agreed
➔ **Changes made:** "described" inserted.

l12: not nice: to an error of 1.5 on τ, of 60hPa on Ptop and of 7.5hPa on ΔP

➔ **Response:**
➔ **Changes made:** suggested text changes made.

l14: Our uncertainty is approximately: What does it mean?

➔ **Response:** This refers to some results from Richardson et al. (2017), we have rephrased.
➔ **Changes made:** p9L29—32 now reads: "Our τ prior error comes from applying the ±18 % error in simulated radiance for homogeneous clouds when provided with MODIS optical depth (Richardson et al., 2017). Our Ptop uncertainty is from the standard deviation of the differences between OCO-2 and CALIPSO P_top when using a simple lookup table for OCO-2, which we intend to use for the OCO-2 prior. The ΔP uncertainty is similar to the ±20 % error associated with Eq. (8) for clouds of cloud fraction > 0.8 reported in (Bennartz, 2007)."

l18-19: 'more intuitive': not very nice, more qualitative ?

➔ **Response:** We feel that either option is ok, but I don't know how to calculate "quantitative-ness" of using the square root of an element of a covariance matrix versus information content. However, we think that most readers will find values expressed in optical depth units or hPa to be more intuitive than information content in bits so prefer to keep the current phrasing.
➔ **Changes made:** None

p9
Description of figure 3: I am confused as the caption seems to say that there are two figures (top and bottom), but only one is visible. Description of figure 4: I don't know where to see the channels you are mentioning (l9) as the plot is in function of the OCO-pixels. It might be a good idea to show a spectra of OCO lines.

➔ **Response:** Figure 3 was changed just prior to submission and caption was not, we've fixed it. Our new Figure 6 contains an OCO-2 spectrum along with an approximated GOME-2 spectrum of the same scene after determining the micro-window to use and calculating information metrics. We show it this late in the paper since showing it before the IC calculations and micro-window selection might confuse readers. To help further we changed the x coordinates of Figure 4 to wavelength and then presenting a spectrum later would be sufficient for readers to follow. The inclusion of a GOME-2-like spectrum and calculation is to help readers understand the extra value of OCO-2's high spectral resolution. This addresses various reviewer comments, including your first one about what value we add and how we compare with previous work.
➔ **Changes made:** Figure 3 caption rewritten, keeping single figure. Figure 4 xlabel changed to wavelength units. New Figure 6 is an example spectrum with the 75 channel micro-window highlighted, a GOME-2 equivalent spectrum and the degrees of freedom for signal added to the legend.

Description of figure 5:
l20 content**2**. remove the 2.

➔ **Response:** Good spot, thanks.
➔ **Changes made:** Deleted the 2.

l23: Again showing a spectra with your selected window might be a good idea.

➔ **Response:**
➔ **Changes made:** See previously, spectrum added in Figure 6.

Also How did you choose the thresholds? You should justify more the choice of 75p as it is not obvious from the plot. 50p could be fine also?

➔ **Response:** We have edited the text to emphasise that we also aimed to consistently satisfy the $P_{top}$ and $\Delta P_c$ criteria as well. The addition of the degrees of freedom for signal to our analysis should also help clarify things.
➔ **Changes made:** p12L16—20 now reads: "The median case in the 50 channel micro-window passes our IC threshold and in all cases passes the τ-uncertainty threshold, but it has multiple cases that fail the P_top and ΔP_c thresholds. By contrast, the 75 channel micro-window containing the OCO-2 channels 353—426 (indices counting from 1 for the full 1,016 OCO-2 L1bSc channels) consistently satisfies our P_top and ΔP_c criteria and reduces the full wavelength range from 759.2—771.8 nm to 763.5—764.6 nm."

p10
l2-3: Once again, showing a spectra would help the reader to follow your conclusions.

➔ **Response:**
➔ **Changes made:** See previously, Figure 6 displays spectrum.

l9-10-11: Sentence too long.

➔ **Response:**

➔ **Changes made:** Sentence broken into two. Similar changes to nearby sentences.

III. Technical corrections
When you quote a paper within a sentence (p2 l3) you shouldn't put the author's name between parentheses. This study goes beyond Richardson et al (2017) by ....

➔ **Response:** This is a reference manager issue.

➔ **Changes made:** Sentence rewritten to avoid parentheses. We will ensure that, if the paper is accepted, we parentheses throughout will be properly handled.

I don't know what is the AMT policy for that but it would be better to centre your equations.

➔ **Response:** We're using the template, it seems AMT formatting changes this.

➔ **Changes made:** None now, will use AMT format if accepted.

In the bibliography, you might think to put the first authors in bold and the titles in Italic; otherwise it is very difficult to distinguish the different papers.

➔ **Response:** We used a reference manager plugin with the template, it seems that, if accepted, AMT has a different format to AMTD which will fix this.

➔ **Changes made:** None now, will use AMT format if accepted.

Figures: In general, be careful with the size of the axis-labels which are very small (fig 2 , 4)

➔ **Response:** Agreed. If accepted we will keep an eye on this make sure that resizing figures doesn't make the text too small.

➔ **Changes made:** Axis label fontsize default increased.

The numbers of the lines restart at 0 at each page, I don't know if it is a mistake or not.

➔ **Response:** We re-downloaded the AMTD template and found the same, it appears to be a template choice.

➔ **Changes made:** None.

[Figure]

**Figure 1 Example simulated cloudy scene A-band spectrum, for a $\tau$ = 10, $P_{top}$ = 850 hPa cloud in a tropical atmosphere with a solar zenith angle of 45°. The black line shows the full OCO-2 simulated spectrum, the blue line is the black line resampled using approximate GOME-2 instrument line shapes and the red line is the selected 75 channel micro-window for OCO-2 cloud retrievals. The legend also reports the $d_s$ for each spectrum with the GOME-2 instrumental uncertainty based on an SNR of 100 as in previous work (Schuessler et al., 2014).**

---

## Author Comment (AC3) · 8 Jan 2018

**General response to reviewer 3**
We have responded to each of your points below, with your text in red and ours in blue. We have made changes to address all of your technical comments. After playing around with the organisation and ordering of the text, we decided to keep the current structure but change section titles to help readers follow.

We have found that we get pretty good match with the MODIS cloud flag by using a simple brightness threshold, and similarly get a good match (~93 % agreement) on cloud phase using a simple derived Nakajima-King diagram. Our retrieval is now running through the OCO-2 data record but we haven't finalised thresholds for warning levels etc. For SZA < 60° We can match when MODIS is confidently cloudy in 93—96 % of cases depending on the orbit, for a set of test orbits we have done. We can also increase the thresholds to do fewer retrievals, but be more likely to retrieve when MODIS also obtains cloud properties (the numbers differ since MODIS retrieval does not provide e.g. $\tau$ for every confidently cloudy scene).

We feel that adding these preliminary results would clutter the paper and the thresholds may change before product release so have just expanded the discussion to address your points.

Your suggestion of adding degrees of freedom for signal and adding a comparison versus lower spectral resolution results has clearly improved the paper. Fortunately we easily replicated the Schuessler et al. GOME-2 results and they fit very neatly into the narrative in our introduction.

Thanks for your time and helpful review.

**NOTE:** our page and line numbers refer to the new version. With our greatly expanded introduction and other minor corrections it became very messy otherwise. New Figure 6 is appended to end of this document.

Review comments on manuscript "Information content of OCO-2 oxygen A-band channels for retrieving marine liquid cloud properties"

Authors: M. Richardson and G. L. Stephens

MS No.: amt-2017-314

MS Type: Research article

General comments:
This paper presents a theoretical study on retrieving marine boundary layer cloud optical thickness, pressure thickness, and top pressure, using the OCO-2 oxygen A-band measurements. The method is well defined and the results are of interests to the community. The topic is suitable for publication in AMT, but I do have some concerns for the authors to consider.

1) Marine boundary layer clouds are targets that we have pretty good a priori knowledge; hence it's not surprising to have good retrieval accuracy, but since the goal of the research is to apply the method to OCO-2 retrievals, one question would be how to decide when to retrieve? I would suggest adding at least some discussions on how to identify the clouds that are suitable for applying this method.

➔ **Response:** We are preparing a paper and Algorithm Theoretical Basis Document to explain our full implementation in practice and didn't want to clutter this paper. However, we understand that readers may be interested in some of the general principles we will apply so have made the changes listed below. Some further justification: the OCO-2 preprocessors provide cloud flags, but they are optimised for glint only over ocean (e.g. Taylor et al. 2016, cited in the text for its

collocated data) and we found we could not use it in nadir. Instead, we use a simple brightness threshold after accounting for solar zenith angle. Initial tests show >90 % agreement with MODIS cloud flags, depending on the orbit.

➔ **Changes made:** Text added to discussion: "Cloud identification is relatively simple for nadir A-band reflectance measurements over ocean, as for most solar zenith angles the surface is dark and cloudy scenes may simply be identified when reflectance exceeds some threshold, which depends on the viewing geometry".

2) The literature review should have been more complete. There have been studies on retrieving cloud pressure thickness plus cloud top pressure in the past, especially for thick clouds over dark surfaces (e.g., Ferlay et al. 2010, Yang et al. 2013, Merlin et al., 2016, reference given below).

➔ **Response:** Agreed, the introduction has been changed.
➔ **Changes made:** The introduction has been rewritten and substantially lengthened with citations to Hanel (1961), Yamamoto & Wark (1961), Deschamps et al. (1994), Ferlay et al. (2010), Desmons et al. (2013), Merlin et al. (2016), Yang et al. (2013), Rozanov & Kokhanovsky (2004), Schuessler et al. (2014), Heidinger & Stephens (2000) and O'Brien & Mitchell (1992). These support a new summary of various A-band cloud studies and then justify our new work as applying hyperspectral approaches that are useful for low clouds. We cite Bony & Dufresne (2005) and Zelinka et al. (2012) to support the importance of low clouds that are poorly sampled by the multi-angular approaches, and explain our advantages for geometrical thickness relative to other work that used instruments with lower SNR and spectral resolution.

3) I found the structure of the paper makes understanding the contents difficult. I would suggest some re-arrangements. For example, Section 2 is titled "The OCO-2 satellite and its instruments", I couldn't see how the two subsections fit there: " 2.1 OCO-2 radiative transfer calculations" and "2.2 Optimal estimation and information content". My suggestion would be to use one section to describe forward modeling issues and another section for retrieval related issues.

➔ **Response:** We looked into moving things around but decided that we prefer the current organisation, but agree that the titles are confusing. Our separation is (Section 2) general information and techniques introduced and (Section 3) Specific techniques and samples used in this paper. Going for (Optimal Estimation) followed by or leading (Forward model) also seemed somewhat confusing given it leaves no obvious place to put the synthetic retrievals, and aspects needed for the optimal estimation (e.g. our sampling methodology) require understanding of the forward modelling and vice versa. We think that whichever way could confuse some readers, but think that after modifying Section 2 and 3 titles things are clearer this way.
➔ **Changes made:** Section 2 title changed to "Data sources and analysis techniques"

4) I would suggest converting the information content shown in the article to how many pieces of information can be retrieved. For example, it's not clear to me what information content = 16 means (the red line in Figure 4(a)) physically.

➔ **Response:** We had thought that the low posterior errors were sufficient to indicate that the degrees of freedom or signal approached 3, but agree that there isn't actually that clear. We have therefore added $d_s$ to our figures and discussion.
➔ **Changes made:** degrees of freedom for signal introduced in Section 3 along with the equation we use to calculate it. Figures 4 and 5 now have it, and legend of new Figure 6 includes example values.

References:
Ferlay, N., and F. Thieuleux, C. Cornet, and A. B. Davis, 2010: Toward New Inferences about Cloud Structures from Multidirectional Measurements in the Oxygen A Band: Middle-of-Cloud Pressure and Cloud Geometrical Thickness from POLDER-3/PARASOL. J. Appl. Meteor. Climatol., 49, 2492–2507. doi: C2 http://dx.doi.org/10.1175/2010JAMC2550.1.
Merlin, G., Riedi, J., Labonnote, L. C., Cornet, C., Davis, A. B., Dubuisson, P., .Parol, F., 2016: Cloud information content analysis of multi-angular measurements in the oxygen A-band: Application to 3MI and MSPI. Atmospheric Measurement Techniques, 9(10), 4977-4995. doi:http://dx.doi.org/10.5194/amt-9-4977-2016.
Yang, Y., A. Marshak, J. Mao, A. Lyapustin, J. Herman, 2013: A Method of Retrieving Cloud Top Height and Cloud Geometrical Thickness with Oxygen A and B bands for the Deep Space Climate Observatory (DSCOVR) Mission: Radiative Transfer Simulations. J. Quant. Spectrosc. Radiat. Trans. 122, 141-149,

http://dx.doi.org/10.1016/j.jqsrt.2012.09.017.

[Figure]

**Figure 1 Example simulated cloudy scene A-band spectrum, for a $\tau = 10$, $P_{top} = 850$ hPa cloud in a tropical atmosphere with a solar zenith angle of 45°. The black line shows the full OCO-2 simulated spectrum, the blue line is the black line resampled using approximate GOME-2 instrument line shapes and the red line is the selected 75 channel micro-window for OCO-2 cloud retrievals. The legend also reports the $d_s$ for each spectrum with the GOME-2 instrumental uncertainty based on an SNR of 100 as in previous work (Schuessler et al., 2014).**

---

## Author Response (AR1)

**Response to reviewers and manuscript with tracked changes for "Information content of OCO-2 oxygen A-band channels for retrieving marine liquid cloud properties"**

Mark Richardson and Graeme L. Stephens

- 5 We have substantially modified our paper in response to reviewer comments. The Introduction has been lengthened and explains how past work suggested that OCO-2's spectral resolution is sufficient to obtain cloud geometric thickness, whereas other instruments (e.g. SCIAMACHY, GOME) were not able to. We also explain how our approach is complementary to multi-angular measurements (e.g. POLDER, MSPI, 3MI) which contain information for physically thicker clouds, and discuss other modern instruments with similar spectral resolution (TROPOMI on Sentinel-5P and the spectrometers on GOSAT & the
- 10 FengYun-3 series).

Our new Figure 6 shows an OCO-2 cloudy scene spectrum with our selected micro-window to help readers contextualise our results, as suggested particularly by reviewer 1, but which addresses issues raised by other reviewers. This figure also includes an example GOME-2-like sampling of the same spectrum and reports the degrees of freedom for signal, showing that OCO-2 contains information on geometric thickness whereas the GOME-2-like signal does not. This fits with our

- 15 As suggested by reviewer 3, we have added degrees of freedom for signal to our information content analysis and show it on the relevant figures. This does not change any of the conclusions but eases interpretation for many readers. A limitation of our work that was not adequately emphasised in the initial submission is overlying aerosol. We now discuss this in more detail in the text, but have not been able to implement proper aerosol-over-cloud calculations, this is under development now.
- 20 We believe that we have responded sufficiently to all reviewer comments and have made changes where necessary. With the link from the new introductory text to the degrees of freedom for OCO-2 versus GOME-2 example, we have a clear explanation of why our retrieval is new and adds to currently available products. We are thankful to the editor and reviewers for their time and attention, which has led to a greatly improved paper.
- 25 Contents:

| 1  | Introduction                    |
|----|---------------------------------|
| 2  | Response to reviewer 1          |
| 9  | Response to reviewer 2          |
| 12 | Response to reviewer 3          |
| 15 | Response to reviewer 4          |
| 20 | Manuscript with tracked changes |

**General response to reviewer 1**

We have responded to each of your points below, with your text in red and ours in blue and believe we have addressed your major concerns. We did not act on some of your minor suggestions but have justified this in each case. Typically this is because of linguistic style choices or because of the AMTD template.

5

The largest changes made in response to your comments are that the introduction has been greatly extended and we have added a new Figure 6. This contains an example OCO-2 spectrum, highlights our micro-window and also shows a GOME-2-like spectrum. These make the paper much more accessible and allow much easier comparison with other instruments.

10

It is obvious that you read our submission with great attention, thank you for your time and feedback.

**NOTE:** our page and line numbers refer to the new version of the manuscript without tracked changes. With our greatly expanded introduction and other minor corrections it became very messy otherwise.

15

**Detailed review on the paper: Information content of OCO-2 oxygen**

A-band channels for retrieving marine liquid cloud properties.

**20**

I. General comments

I think this paper is very interesting and brings innovation on how to retrieve cloud properties with OCO-2. The use of optimal estimation method makes the study very robust.

- 25 I have some remarks concerning the introduction. I think you should rework it to make it more complete. Indeed you should answer the following questions:
  - What are the motivations for this study?
  - What has already been done?
  - What does your study bring?

As those aspects are not clear. I also find your bibliography too light. We don't expect you to quote all the works done in the O2 A-band and optimal estimation, but at least some of them. You can read the paper of Merlin et al (2017) as the subject is close to yours and the bibliography is quite complete.

35

30

→ Response: We tried to keep the paper concise, but now agree that we missed too much context so have made major changes.

→ Changes made: Much rewritten and added text, covering p1L18—p3L34. The introduction has been rewritten and lengthened with citations to Hanel (1961), Yamamoto & Wark (1961), Deschamps et al. (1994), Ferlay et

- 40 al. (2010), Desmons et al. (2013), Merlin et al. (2016), Yang et al. (2013), Rozanov & Kokhanovsky (2004), Schuessler et al. (2014), Heidinger & Stephens (2000) and O'Brien & Mitchell (1992). These support a new summary of various A-band cloud studies and then justify our new work as applying hyperspectral approaches that are useful for low clouds. We cite Bony & Dufresne (2005) and Zelinka et al. (2012) to support the importance of low clouds that are poorly sampled by the multi-angular approaches, and explain our
- 45 advantages for geometrical thickness relative to other work that used instruments with lower SNR and spectral resolution.

**II. Specific comments**

50

p1

L 19-20, there are numerous papers that you can quote.

- → Response: See changes above.
- → Changes made: Introduction fully rewritten.

**5 p2**

I25: multiply scatter : not nice

- **> Response:** Term deleted, the lidar being attenuated justifies the point on its own.
- → Changes made: "...attenuate and multiply scatter the CALIPSO lidar" → "attenuate the CALIPSO lidar"

**10**

I25-26-27-28: This sentence is too long

- → Response: Agreed.
- → Changes made: Sentence split into two.
- 15 I31: This work ....: Sentence not clear
  - → **Response:** Justification added.
  - → Changes made: Sentence now reads: "Since any footprint that is identified as possibly cloudy is not processed in the standard OCO-2 products this work generates value from largely unused soundings."

**20**

25

р3

14: do contain information.... Reference is missing

- → Response: This is based on Nakajima-King-like principles but I don't have the formal information content analysis for the OCO-2 instrument. Therefore we changed the wording slightly and added a citation.
- → Changes made: p4L22—25 changed and now reads "The CO2 bands are not considered in this analysis but do inform about cloud phase and droplet or particle size (Nakajima and King, 1990), and this information will be used when this retrieval is applied in our observation-based study to identify likely liquid cloud cases."
- 30 I21 ECMWF meteorological fields : Reference missing

**➔ Response:**

→ Changes made: p5L10—11 added text: "response as described in the OCO-2 data version 6 documentation (Boesch et al., 2015)"

**35**

p4

118 observed and expected y : is a value missing after "observed"?

- → Response: The meaning is intended as "observed y and expected y" but that feels clunky to me. Another option is to hyphenate to "observed- and expected y", but grammar guides now disagree over that use and it seems archaic. I thought context made it clear, but have added a little extra text to further emphasise the context.
  - Changes made: p6L14—15 rewritten slightly to: "based on the difference between the observed and expected y"
- 45

50

40

115 to 30: When you refer to a vector or a value you could write its symbol

- → Response: Symbols added to aid the reader, with minor rephrasing so that it's clear that S-hat refers to the posterior uncertainty and not the "reduction in posterior uncertainty".
- → Changes made: Vector and matrix symbols added and text changed, e.g. "reduction in posterior uncertainty"
   → "posterior uncertainty \$\hfrac{1}{5}\$ is reduced by..."

I22 observation vector instead of observation state vector

- → Response:
- → Changes made: change made.

5

I22 a point is missing after channels

- → Response:
- → Changes made: change made.
- 10

127 Shannon entropy : Reference missing

- → Response:
- → Changes made: p6L30 now reads "...and this change in associated Shannon entropy (Shannon and Weaver, 1949)..."
- p5

11: You don't define P0 and P1

20

15

- ➔ Response:
- → Changes made: p7L2—4 now reads "In this case  $S(P_0)$  is the Shannon entropy associated with the original probability distribution and  $S(P_0)$  the same value associated with the retrieved probability distribution."
- 25 I6 :see my comment p4 I15
  - → Response:
  - → Changes made: Symbol added.
- 30 I19 : Methodology **and** example atmosphere **and** cloud .. Not nice.
  - → Response:
  - → Changes made: Changed to "Methodology, atmospheric states and cloud cases"
- 35
- p6

i1 pw not present in eq 8

- 40  $\rightarrow$  **Response:** Good catch, this was a typo.
  - → Changes made: \rho converted to \rho\_w in Equation 8.

I7: Why do you take Qext =2?

- 45 **Response:** Size parameters  $x = 2\pi r/\lambda$  here are >50 and water is weakly absorbing (real part of index ~1.33, imaginary part ~1×10-7), so I take  $lim_{x\to\infty}$  case for a non-absorbing sphere.
  - Changes made: p8L6—7 text added: "This value is chosen as it represents the large-particle limit for nonabsorbing spheres (Herman, 1962) which is a reasonable approximation for cloud droplets in the oxygen Aband"
- 50

17: 0°-20°, 20°-50° and 50°-90°, you forgot the degree symbol over 0, 20 and 50.

- → Response: This appears to be an AMT style choice. Under "English guidelines and house standards" it says "En dashes (-) are longer than hyphens (-) and serve numerous purposes....En dashes are used to indicate, among other things, relationships (e.g. ocean-atmosphere exchange), ranges (e.g. 12–20 months),..." this implies that for ranges the unit follows the latter value only.
- 5 **Changes made:** None

17: 'identified as single-layer liquid clouds by both MODIS and CaLiPSO'. It may be useful for the reader to write which product/ collection you used.

10

**→ Response:**

- → Changes made: p8L14—15 now reads: "The MODIS data are from product MYD06 at 1 km horizontal resolution (Platnick et al., 2015) and the CALIPSO data are from the 1 km resolution cloud layer product 01kmCLay (Vaughan et al., 2009)."
- 15

18-9: You should rewrite the 2 sentences which are not clear. For instance :

'Within each bin, all the OCO-2 ECMWF-Aux profiles (including pressure, temperature, humidity and wind speed) are averaged level by level.'

- 20
- → **Response:** Agreed.
- → Changes made: p8L15—17 now use your suggested text.

I22: not nice. You should rewrite the description of the uncertainties, particularly for the humidity.

25

30

- → Response: The humidity method description was split by the temperature sampling description, we've rewritten to ensure that the specific humidity perturbations are described continuously and hope that this is clearer.
- Changes made: p8L30—p9L2 now reads: "For temperature we add a uniform perturbation to each level with a value sampled from a zero mean (μ) Gaussian with standard deviation (σ) of ±1.5 K. For specific humidity we sample from a zero mean Gaussian with a standard deviation of unity, then scale this value based on pressure level. The scaling is equivalent to ±20 % of the initial specific humidity at the surface, increasing linearly to ±50 % of the layer values at 250 hPa and remaining at ±50 % for levels with lower pressure."
- 35 I25: standard deviation **of** +-1.5K we sample: what are you sampling?
  - → **Response:** Above text change hopefully addresses this.
  - → Changes made: See above.
- 40

I26: with 2000 perturbations applied to reff

- ➔ Response.
- → Changes made: "applied" added.
- 45

I27: '5--95% range of 7.5--19.4 um' Not sure of what it means. Try to avoid the abbreviations in the text and write a sentence.

- 50
- → **Response:** We have rewritten this in a way that we hope is clearer.
- → Changes made: p9L4—5 now reads: "This lognormal fit has an arithmetic mean of 12.0 µm, but after excluding values outside the 4—30 µm retrieved by MODIS, the arithmetic mean is 12.6 µm and 5—95 % of the values fall within 7.5—19.4 µm."

129: The output was sampled: You are using this word quite often and maybe not always with the right sense?

- → **Response:** Agreed.
- → Changes made: p9L12—13 now reads: " The output spectra are calculated for each of the 8 different
- instrument line shapes associated with the 8 different OCO-2 across-track sounding positions"

5

p7 I8: cases **described** in sect. 3.1

- 10
- → Response: Agreed
- → Changes made: "described" inserted.

I12: not nice: to an error of 1.5 on  $\tau$ , of 60hPa on Ptop and of 7.5hPa on  $\Delta P$

- 15
- → Response:
- → Changes made: suggested text changes made.
- I14: Our uncertainty is approximately: What does it mean?
- 20
- → **Response:** This refers to some results from Richardson et al. (2017), we have rephrased.
- Changes made: p9L29—32 now reads: "Our T prior error comes from applying the ±18 % error in simulated radiance for homogeneous clouds when provided with MODIS optical depth (Richardson et al., 2017). Our Ptop uncertainty is from the standard deviation of the differences between OCO-2 and CALIPSO P\_top when
- 25 using a simple lookup table for OCO-2, which we intend to use for the OCO-2 prior. The  $\Delta P$  uncertainty is similar to the ±20 % error associated with Eq. (8) for clouds of cloud fraction > 0.8 reported in (Bennartz, 2007)."
  - I18-19: 'more intuitive': not very nice, more qualitative ?
- 30
- → Response: We feel that either option is ok, but I don't know how to calculate "quantitative-ness" of using the square root of an element of a covariance matrix versus information content. However, we think that most readers will find values expressed in optical depth units or hPa to be more intuitive than information content in bits so prefer to keep the current phrasing.
- 35 **Changes made:** None

**p9**

40

45

Description of figure 3: I am confused as the caption seems to say that there are two figures (top and bottom), but only one is visible. Description of figure 4: I don't know where to see the channels you are mentioning (I9) as the plot is in function of the OCO-pixels. It might be a good idea to show a spectra of OCO lines.

- → Response: Figure 3 was changed just prior to submission and caption was not, we've fixed it. Our new Figure 6 contains an OCO-2 spectrum along with an approximated GOME-2 spectrum of the same scene after determining the micro-window to use and calculating information metrics. We show it this late in the paper since showing it before the IC calculations and micro-window selection might confuse readers. To help further we changed the x coordinates of Figure 4 to wavelength and then presenting a spectrum later would be sufficient for readers to follow. The inclusion of a GOME-2-like spectrum and calculation is to help readers understand the extra value of OCO-2's high spectral resolution. This addresses various reviewer comments, including your first one about what value we add and how we compare with previous work.
- 50 → Changes made: Figure 3 caption rewritten, keeping single figure. Figure 4 xlabel changed to wavelength units. New Figure 6 is an example spectrum with the 75 channel micro-window highlighted, a GOME-2 equivalent spectrum and the degrees of freedom for signal added to the legend.

Description of figure 5: 120 content**2**. remove the 2.

- → **Response:** Good spot, thanks.
  - → Changes made: Deleted the 2.

I23: Again showing a spectra with your selected window might be a good idea.

10

5

- → Response:
- → Changes made: See previously, spectrum added in Figure 6.
- 15 Also How did you choose the thresholds? You should justify more the choice of 75p as it is not obvious from the plot. 50p could be fine also?
  - → **Response:** We have edited the text to emphasise that we also aimed to consistently satisfy the  $P_{top}$  and  $\Delta P_c$  criteria as well. The addition of the degrees of freedom for signal to our analysis should also help clarify things.
- 20 → Changes made: p12L16—20 now reads: "The median case in the 50 channel micro-window passes our IC threshold and in all cases passes the τ-uncertainty threshold, but it has multiple cases that fail the P\_top and ΔP\_c thresholds. By contrast, the 75 channel micro-window containing the OCO-2 channels 353—426 (indices counting from 1 for the full 1,016 OCO-2 L1bSc channels) consistently satisfies our P\_top and ΔP\_c criteria and reduces the full wavelength range from 759.2—771.8 nm to 763.5—764.6 nm."

**25**

p10

12-3: Once again, showing a spectra would help the reader to follow your conclusions.

- → Response:
- 30 → Changes made: See previously, Figure 6 displays spectrum.

I9-10-11: Sentence too long.

- → Response:
- **35 Changes made:** Sentence broken into two. Similar changes to nearby sentences.

**III. Technical corrections**

When you quote a paper within a sentence (p2 l3) you shouldn't put the author's name between parentheses. This study goes beyond Richardson et al (2017) by ....

- → **Response:** This is a reference manager issue.
- → Changes made: Sentence rewritten to avoid parentheses. We will ensure that, if the paper is accepted, we parentheses throughout will be properly handled.

**45**

I don't know what is the AMT policy for that but it would be better to centre your equations.

- → **Response:** We're using the template, it seems AMT formatting changes this.
- → Changes made: None now, will use AMT format if accepted.

**50**

In the bibliography, you might think to put the first authors in bold and the titles in Italic; otherwise it is very difficult to distinguish the different papers.

- → Response: We used a reference manager plugin with the template, it seems that, if accepted, AMT has a different format to AMTD which will fix this.
- → Changes made: None now, will use AMT format if accepted.
- 5 Figures: In general, be careful with the size of the axis-labels which are very small (fig 2, 4)
  - → Response: Agreed. If accepted we will keep an eye on this make sure that resizing figures doesn't make the text too small.
  - → Changes made: Axis label fontsize default increased.

10

The numbers of the lines restart at 0 at each page, I don't know if it is a mistake or not.

- **Response:** We re-downloaded the AMTD template and found the same, it appears to be a template choice.
- → Changes made: None.

15

**General response to reviewer 2**

We have responded to each of your points below, with your text in red and ours in blue.

Firstly, we haven't specifically mentioned airborne data, although some of our citations and text now refer to instruments that have airborne versions (e.g. MSPI). We're definitely interested in seeing the outcome of more airborne campaigns and comparing our functioning retrieval with available airborne data including non-A-band sensors. A next step is to look at ORACLES data to assist with our validation since they have some flights designed to underpass CloudsSat, which has the same reference ground track as OCO-2

- 10 The ORACLES example brings us neatly to the issue of aerosols: our current radiative transfer implementation has had some problems with adding above cloud aerosol. We have plans to transfer the code to optimise for scattering atmospheres but we believe that sufficient caveats mean that this paper is still justified (after all, other recent cloudy information content papers have worked on single layer cloudy scenes too!). We have added text and citations regarding aerosols and highlight that it is a source of uncertainty that we must address.
- 15

Your suggestion of using the multi-layer mask from MODIS is excellent and we are considering and testing it now. We are currently running our retrieval with CALIPSO priors as well, but MODIS has the advantage of a longer expected time in the A-train. Ultimately we would like to identify multi-layer cases with OCO instrumentation alone so that our retrieval could be applied to e.g. potential OCO-3 measurements even if no other MODIS-like or CATS-like instruments are currently be tableted with the advantage of th

20 available to identify multi-layer cases. However, CloudSat-CALIPSO and MODIS multi-layer data are vital to allow us to develop and test this technique.

Thanks for taking the time to review our paper, you spotted several unclear points or typos that we have now fixed.

25 **NOTE:** our page and line numbers refer to the new version of the manuscript without tracked changes. With our greatly expanded introduction and other minor corrections it became very messy otherwise.

Review of Richardson & Stephens paper:

- 30 This is a very interesting and valuable study. I would be very interested to know how this study could transfer to airborne spectrometers like AVIRIS and PICARD that also have high spectral resolution and lack IR channels for cloud top retrieval. We've done a similar thing with ASTER: used an instrument that was previously only for clear-sky work and created a product from unused data. The paper is overall well written and methods are clearly described and understandable.
- 35

**Major comments:**

Marine SCu frequently have some kind of aerosol sitting on top of them especially off the coast of Africa (Sahara dust and Namibia smoke) and to a lesser extent the US Pacific Coast (mostly smoke). Have you tried inserting above-cloud aerosol layers into your simulations and seeing what happens? I'm not saying

- 40 that you have to correct for aerosols but some idea as to uncertainty introduced by absorbing aerosols would be nice.
  - → Response: We have added some discussion about aerosols and indicated that we do not consider them in this study. There have been some technical problems implementing aerosol layers into our modified cloudy-scene radiative transfer model. Much other A-band work has considered clear-sky cases and we have
- 45 discussed the prevalence of aerosol in the new text, and note that we should be able to flag heavily polluted cases using collocated data. Future work will look in more detail at overlying aerosol, and speculatively I expect an effect on the residual spectral fits from the retrieval which may allow identification based on OCO-2 alone.

- Changes made: p3L10—12 now reads: "Our current analysis considers aerosol-free cases as aerosols have not yet been properly implemented in our modified cloudy-sky version of the radiative transfer model, this is an avenue for future work and will be discussed in Sect. 5."
- P13L32—p14L9 now reads: "Alternatively, since OCO-2 flies in the A-train it would also be possible to use other sensors such as CALIPSO (which is now leaving the A-train) or MODIS to identify multi-layer cloud cases, or scenes in which there is heavy aerosol loading. Cases of heavy aerosol loading are most common over the Namibian stratocumulus region with common occurrence in June-July-August (JJA) and a peak in September-October-November (SON). A combination of CALIPSO, CloudSat and International Satellite Cloud Climatology Project (ISCCP) data imply that in the SON Namibian stratocumulus region, approximately one-third of low clouds have overlying aerosol, and approximately half of these cases are smoke (Devasthale and Thomas, 2011; Winker et al., 2010). Scattering layers overlying a marine cloud tend to reduce in the effective retrieved cloud layer pressure due to the reduced mean path length of those photons reflected from the overlying layer (Vanbauce et al., 1998). Assessment of aerosol effects will be necessary in future work."

15

Please be consistent in definition of micro-window. You use "pixels" in the first 8.5 pages of the paper and then switch to "channels" for the rest of the text. I personally would prefer you use "channels", but you can use whichever you see fit as long as it's consistent throughout.

- Response: Agreed, this was a legacy from our use in a previous paper and some OCO-2 documentation but is confusing.
  - → Changes made: We now use the correct term "focal plane array elements" when discussing damage to the sensor, and "channels" for all spectral properties.

**25**

20

**Minor comments:**

Figure 3 caption should read  $\mu_{0-2}=\cos_{-2}(SZA)$ ,  $\mu$  is normally used to indicate sensor zenith angle.

- → Response: Oops.
- → Changes made: Labels changed throughout,  $\mu \rightarrow \mu_0$

**30**

Page 1 Line 1: please expand CALIPSO acronym, first use

→ Response: Done.

acronym is expanded.

→ Changes made: CALIPSO is now introduced on p2L13 following our major changes to the introduction, its

35

- Page 2 Line 21: should read "equator crossing time near 13:30"
  - → Response:
- 40 **→ Changes made:** Done.

Page 7 Line 25: please clarify what the micro-windows are measured in: 500 of what? Later in the text, on page 9 it becomes clear that the units of the micro-window size are channels. For folks that don't

45 normally use something like OCO, it might help giving a bit more information, like what a 75-channel micro-window translates into as far as a wavelength range goes. It would make the research more transferable to other instruments as this is a potentially very valuable retrieval approach.

- → Response: This was unclear on our part. We have now clarified throughout, adding "channels" after 500. The 75 channel wavelength range is given. The new Figure 6, made in response to reviewer 1, also hopefully clarifies things.
- → Changes made: p10L9—10 now reads: "To make this problem tractable, we select micro-windows of the following size: 5, 10, 25, 50, 75, 100, 150, 200 and 500 neighbouring channels." (note added word "channels")

p12L17—28 text includes: "By contrast, the 75 channel micro-window...reduces the full wavelength range from 759.2—771.8 nm to 763.5—764.6 nm."

10 Figure 6 added to visualise this.

Page 9 Line 3: please use  $\theta_0$  and  $\mu_0$  as is generally customary for solar zenith angle and its cosine

- → **Response:** Agreed.
- 15 → Changes made: Done.

Page 9 Line 20: "highest mean information content2." A typo?

- → Response: Good catch.
- 20 → Changes made: 2 deleted.

Page 10 Line 26: OCO is in the constellation with Aqua, so you may be able to use the MODIS multilayer cloud map in order to stay away from cirrus. That's just what that map is for.

- 25 → **Response:** We have done some preliminary analysis using this for validation of the retrieval and it makes a difference (same as multiple layers from CALIPSO). This was helpful in our discussion, thanks.
  - → Changes made: See aerosol response text, we now mention the multi-layer map.

30

**General response to reviewer 3**

We have responded to each of your points below, with your text in red and ours in blue. We have made changes to address all of your technical comments. After playing around with the organisation and ordering of the text, we decided to keep the current structure but change section titles to help readers follow.

5

We have found that we get pretty good match with the MODIS cloud flag by using a simple brightness threshold, and similarly get a good match (~93 % agreement) on cloud phase using a simple derived Nakajima-King diagram. Our retrieval is now running through the OCO-2 data record but we haven't finalised thresholds for warning levels etc. For SZA < 60° We can match when MODIS is confidently cloudy in 93—96 % of cases depending on the orbit, for a set of text arbits we have done. We can get a done when models to do fewer retrievals but we have a thresholds to do fewer retrievals but we have a set of the match when models is confidently cloudy in 93—96 % of cases depending on the orbit, for a set of text arbits we have a done.

10 test orbits we have done. We can also increase the thresholds to do fewer retrievals, but be more likely to retrieve when MODIS also obtains cloud properties (the numbers differ since MODIS retrieval does not provide e.g.  $\tau$  for every confidently cloudy scene).

We feel that adding these preliminary results would clutter the paper and the thresholds may change before product release so have just expanded the discussion to address your points.

Your suggestion of adding degrees of freedom for signal and adding a comparison versus lower spectral resolution results has clearly improved the paper. Fortunately we easily replicated the Schuessler et al. GOME-2 results and they fit very neatly into the narrative in our introduction.

20

Thanks for your time and helpful review.

**NOTE:** our page and line numbers refer to the new version of the manuscript without tracked changes. With our greatly expanded introduction and other minor corrections it became very messy otherwise.

Review comments on manuscript "Information content of OCO-2 oxygen A-band channels for retrieving marine liquid cloud properties"

Authors: M. Richardson and G. L. Stephens

MS No.: amt-2017-314

40

MS Type: Research article

General comments:

This paper presents a theoretical study on retrieving marine boundary layer cloud optical thickness, pressure thickness,
 and top pressure, using the OCO-2 oxygen A-band measurements. The method is well defined and the results are of interests to the community. The topic is suitable for publication in AMT, but I do have some concerns for the authors to consider.

1) Marine boundary layer clouds are targets that we have pretty good a priori knowledge; hence it's not surprising to have good retrieval accuracy, but since the goal of the research is to apply the method to OCO-2 retrievals, one question would

be how to decide when to retrieve? I would suggest adding at least some discussions on how to identify the clouds that are suitable for applying this method.

- → Response: We are preparing a paper and Algorithm Theoretical Basis Document to explain our full implementation in practice and didn't want to clutter this paper. However, we understand that readers may be interested in some of the general principles we will apply so have made the changes listed below. Some further justification: the OCO-2 preprocessors provide cloud flags, but they are optimised for glint only over ocean (e.g. Taylor et al. 2016, cited in the text for its collocated data) and we found we could not use it in nadir. Instead, we use a simple brightness threshold after accounting for solar zenith angle. Initial tests show >90 % agreement with MODIS cloud flags, depending on the orbit.
- 10 → Changes made: Text added to discussion: "Cloud identification is relatively simple for nadir A-band reflectance measurements over ocean, as for most solar zenith angles the surface is dark and cloudy scenes may simply be identified when reflectance exceeds some threshold, which depends on the viewing geometry".
- 15 2) The literature review should have been more complete. There have been studies on retrieving cloud pressure thickness plus cloud top pressure in the past, especially for thick clouds over dark surfaces (e.g., Ferlay et al. 2010, Yang et al. 2013, Merlin et al., 2016, reference given below).
  - → **Response:** Agreed, the introduction has been changed.
- Changes made: The introduction has been rewritten and substantially lengthened with citations to Hanel (1961), Yamamoto & Wark (1961), Deschamps et al. (1994), Ferlay et al. (2010), Desmons et al. (2013), Merlin et al. (2016), Yang et al. (2013), Rozanov & Kokhanovsky (2004), Schuessler et al. (2014), Heidinger & Stephens (2000) and O'Brien & Mitchell (1992). These support a new summary of various A-band cloud studies and then justify our new work as applying hyperspectral approaches that are useful for low clouds. We cite Bony & Dufresne (2005) and Zelinka et al. (2012) to support the importance of low clouds that are poorly complete by the multi-approaches, and explain our adventages for geometrical thickness relative to the superior of the support of the support of the support of the superior of the superior of the superior of the support of the support of the support of the support of the superior of the superior of the superior of the support of the support of the support of the superior o
- 25 sampled by the multi-angular approaches, and explain our advantages for geometrical thickness relative to other work that used instruments with lower SNR and spectral resolution.
- 3) I found the structure of the paper makes understanding the contents difficult. I would suggest some re-arrangements.
   30 For example, Section 2 is titled "The OCO-2 satellite and its instruments", I couldn't see how the two subsections fit there:
- " 2.1 OCO-2 radiative transfer calculations" and "2.2 Optimal estimation and information content". My suggestion would be to use one section to describe forward modeling issues and another section for retrieval related issues.
- Response: We looked into moving things around but decided that we prefer the current organisation, but agree that the titles are confusing. Our separation is (Section 2) general information and techniques introduced and (Section 3) Specific techniques and samples used in this paper. Going for (Optimal Estimation) followed by or leading (Forward model) also seemed somewhat confusing given it leaves no obvious place to put the synthetic retrievals, and aspects needed for the optimal estimation (e.g. our sampling methodology) require understanding of the forward modelling and vice versa. We think that whichever way could confuse some readers, but think that after modifying Section 2 and 3 titles things are clearer this way.
  - → Changes made: Section 2 title changed to "Data sources and analysis techniques"
- 4) I would suggest converting the information content shown in the article to how many pieces of information can be 45 retrieved. For example, it's not clear to me what information content = 16 means (the red line in Figure 4(a)) physically.
  - → Response: We had thought that the low posterior errors were sufficient to indicate that the degrees of freedom or signal approached 3, but agree that there isn't actually that clear. We have therefore added ds to our figures and discussion.
  - → Changes made: degrees of freedom for signal introduced in Section 3 along with the equation we use to calculate it. Figures 4 and 5 now have it, and legend of new Figure 6 includes example values.

| 1 | ~        |
|---|----------|
| н | ٦ |
|   | ~        |

20

- 25
- 30
- 35
- 40
- 45

**General response to reviewer 4**

We have responded to each of your points below, with your text in red and ours in blue.

- 5 You identified a number of areas where we did not provide enough information for easy replication, and also a number of areas where we did not provide enough context or emphasise key points of interest for more general readers. Our changes have addressed all of your points and your incisive comments greatly improved the paper. Thank you for your time.
- 10 In some cases we struggled between clarity and providing details but have found a balance. For example, we use 12  $\mu$ m for  $r_{eff}$  based on the mean of the full PDF that is fit to MODIS data being 12.0  $\mu$ m, but after applying thresholds the mean shifts to 12.6  $\mu$ m, explaining some apparent differences in our original submission. We have added explanatory text where necessary.
- Our updated introduction and Figure 6 combine to tell a nice story: OCO-2 has sufficient spectral resolution to retrieve geometric thickness, given our assumptions and based on previous work which we now describe in more detail. MERIS or even GOME-2 do not, and our new Figure 6 shows a GOME-2-like calculation where we replicate previous work by Schuessler et al. Multi-angular measurements should retrieve thickness, but only for geometrically thick clouds (multi-km+), so OCO-2 is complementary to them.
- 20

Other instruments have high spectral resolution so the results may carry over, and we list those of which we are aware (TROPOMI on SentineI-5P, the spectrometers on the FengYun-3 series and GOSAT). We also point out where spatial resolutions differ (OCO-2 is better than TROPOMI/GOSAT), but also that OCO-2's narrow swath is a limitation compared with TROPOMI's much better coverage.

25

Changes made based on your comments, and those of the other reviewers, have made us provide much more context and resulted in a richer narrative that we hope will make the paper far easier to read and more widely useful. Thanks once again.

30 **NOTE:** our page and line numbers refer to the new version of the manuscript without tracked changes. With our greatly expanded introduction and other minor corrections it became very messy otherwise.

This paper analyzed the information content in O2 A band for retrieving marine liquid cloud properties. it used the Rodgers (2000) formal optimization framework and expressed the information content in terms of degree of freedom for signal and Shannon entropy. The O2 A band on OCO-2 has 800+ channels, and this paper shows that only \_75 channels are

45 needed to retain all information content for retrieving cloud optical depth, cloud pressure thickness, and cloud-top pressure. The method in this paper is sound, but revisions are needed to include various advances in recent studies of using O2 A and B for cloud/aerosol height retrievals, as well as more justification about assumptions and caveats in this study.

1. Abstract. what is cloud-pressure thickness? what is the unit here?

**➔ Response:**

5

10

20

40

Changes made: Abstract changed: "...and cloud-pressure thickness, which is the geometric thickness expressed in hPa."

2. Introduction. Most references cited in the first paragraph are theoretic work done in the past. While they are interesting, there are renewed interests in recent years to use O2 A and B band to retrieve cloud/aerosol height, with some using real data with good validations. They should be included in this paper, and discussion should be made that recent studies with real data use only O2 A/B bands from an imager (such as EPIC or MERES), although some studies did recommend the use of spectra to retrieve aerosol height. See references below and references therein (including some work done by authors' colleagues in JPL).

- Ding, S. et al., 2016, Polarimetric remote sensing in O2 A and B bands: Sensitivity study and information content analysis for vertical profile of aerosols, Atmospheric Measurement Techniques, 9, 2077-2092.
- Xu, X. et al., 2017, Passive remote sensing of altitude and optical depth of dust plumes using the oxygen A and B bands: First results from EPIC/DSCOVR at Lagrange-1 point, Geophys. Res. Lett., 44, 7544-7554.
  - → Response: We chose to be brief and limit discussion to things relevant to our hyperspectral result, but agree that we sliced out too much contextual work. Our changes were made to address all reviewers and we decided to focus on the cloud-relevant components. We reference Yang et al. (2013) for EPIC/DSCOVR since they looked at clouds. We add Ding et al. later in the text (see comment 5).
- Changes made: The introduction has been rewritten and substantially lengthened with citations to Hanel (1961), Yamamoto & Wark (1961), Deschamps et al. (1994), Ferlay et al. (2010), Desmons et al. (2013), Merlin et al. (2016), Yang et al. (2013), Rozanov & Kokhanovsky (2004), Schuessler et al. (2014), Heidinger & Stephens (2000) and O'Brien & Mitchell (1992). These support a new summary of various A-band cloud studies and then justify our new work as applying hyperspectral approaches that are useful for low clouds. We cite Bony & Dufresne (2005) and Zelinka et al. (2012) to support the importance of low clouds that are poorly sampled by the multi-angular approaches, and explain our advantages for geometrical thickness relative to other work that used instruments with lower SNR and spectral resolution.
- Section 2. It should be made clear if cloud properties are well characterized, will CO2 be retrieved accurately in cloudysky conditions? If so, is it column CO2 above cloud top or whole atmospheric column, including CO2 within cloud? Any references will be helpful in this regard. To what degree of accuracy of cloud properties are needed in order to retrieve CO2 with good accuracy?
  - → Response: Our current retrieval will not allow this because we do not retrieve droplet effective radius, and droplet radius is needed since the wavelength differences between the O2 band and CO2 bands is large enough that it matters for the XCO2 retrieval. We now cite a paper whose figure shows that not knowing the droplet size causes a ~15 ppm spread in above-cloud CO2 retrievals, which is far larger than the <1 ppm that are reported as the requirements for the flux modellers.
  - → Changes made: p3L30—p3L24 now reads: "This approach aims to optimise a cloud property retrieval and due to limitations related to the radiative transfer implementation and computational burden, droplet size is not a retrieved property but contributes to the posterior uncertainty. Above-cloud CO2 retrievals have been found
- 45 to require cloud droplet size for good accuracy (Vidot et al., 2009) and therefore our current implementation will not directly lead to above-cloud CO2 retrievals."

4. Section 2.1. It is noted that there are often aerosol layer above marine boundary layer cloud. To be clear, no aerosol 50 effects are treated in L2RTM, correct? How about surface reflectance?

→ Response: We have had some issues properly integrating aerosols into the cloudy-scene L2RTM, although the operational "clear sky" XCO2 retrieval run for the OCO-2 mission allows for optically thin layers of

stratospheric aerosols. We have added text to emphasise the limitations and that this is future work. We also realise that the surface requires further explanation. We now describe our surface model.

- → Changes made: Regarding the surface, p3L32—p4L2 now reads: "Water surfaces at nadir are dark, and even in cloud-free cases there is rarely sufficient SNR for the OCO-2 algorithm to attempt an XCO2 retrieval. We assume a Cox-Munk surface reflectance function with the L2RTM surface reflectance set to 0.10, but as we only use nadir view over ocean there is little sensitivity to surface properties."
- → Regarding aerosols, p3L10—12 now reads: "Our current analysis considers aerosol-free cases as aerosols have not yet been properly implemented in our modified cloudy-sky version of the radiative transfer model, this is an avenue for future work and will be discussed in Sect. 5."

P13L32—p14L9 now reads: "Alternatively, since OCO-2 flies in the A-train it would also be possible to use other sensors such as CALIPSO (which is now leaving the A-train) or MODIS to identify multi-layer cloud cases, or scenes in which there is heavy aerosol loading. Cases of heavy aerosol loading are most common over the Namibian stratocumulus region with common occurrence in June-July-August (JJA) and a peak in September-October-November (SON). A combination of CALIPSO, CloudSat and International Satellite Cloud Climatology Project (ISCCP) data imply that in the SON Namibian stratocumulus region, approximately one-third of low clouds have overlying aerosol, and approximately half of these cases are smoke (Devasthale and Thomas, 2011; Winker et al., 2010). Scattering layers overlying a marine cloud tend to reduce in the effective retrieved cloud layer pressure due to the reduced mean path length of those photons reflected from the overlying layer (Vanbauce et al., 1998). Assessment of aerosol effects will be necessary in future work."

25 section 2.2. what is the state vector? optical depth, top pressure/height? be clear here. This also applies to the title of this paper. what properties to be retrieved? droplet size, top pressure/height or optical depth?

**➔ Response:**

5

10

15

20

30

40

- Changes made: p6L4—p6L6 now reads: "We follow the principles of optimal estimation from (Rodgers, 2000), where a Bayesian retrieval combines an observation vector y with a prior state vector x\_a and obtains a posterior state x^. In our case the state vector consists of cloud-top pressure P\_top, cloud pressure thickness ΔP\_c and cloud optical depth T."
- 35 5. Page 4, Line 25. Ding et al. (2016) used similar method to select channels needed in O2 A and B band for aerosol retrievals. It is worthy to mention here.

**➔ Response:**

- → Changes made: Text added: "...(Chang et al., 2017; Mahfouf et al., 2015; Martinet et al., 2014; Rabier et al.,
- 2002), and this approach has already been used in an oxygen A-band and B-band analysis for aerosol retrievals (Ding et al., 2016)."

6. Page 5, L20. How about effective variance of cloud droplet size? Does it matter?

45 → Response: See response to comment 8.
→ Changes made:

7. Page 6, L2. "A pressure scale height of 8 km is assumed to convert the resultant....". This sentence is hard to comprehend.

**➔ Response:**

→ Changes made: text re-written as "Cloud geometric thickness is converted to pressure thickness by assuming that pressure decreases exponentially with altitude with a scale height of 8 km"

8. Page 6, L27. mean of 12.6 um? should it be 12 um to be consistent with previously stated? How about effective 5 variance?

- → Response: With our modified L2FP code we have to select integer values of reff and picked a value close to the peak (the exact value depends on the statistic you use: arithmetic mean is 12.6 microns but you can get other values from a log fit, taking the median, mode or others). We have mentioned the integer sampling earlier in the text, but think that adding extra text here is clunky so have made no changes. We have, however, added clarification text where we think it does not interrupt the flow.
- → Changes made: p5L29—p5L32 now reads: "Mie scattering computations are used within louds using relevant coefficients that are pre-calculated for gamma distributions of cloud droplets based on a summary of low-cloud studies (Miles et al., 2000). These values have only been pre-computed for integer values of effective droplet size. This should not affect our results greatly since our calculated uncertainties include a term spanning a range of droplet sizes."

P11L4—o117 now reads: "The r\_eff distribution effective variance is fixed in each case in order to use the precalculated scattering properties used with the L2RTM code, but given the wide range of effective mean values considered, it is not expected that allowing the effective variance to change would greatly affect the results."

9. Page 6, L29. cloud top pressure of 850 hpa? but, in the 3.1, it says three different pressures.

- → Response: We calculated covariance calculations for a single cloud-top pressure but used multiple cloud-top pressures in the calculation of information content and in the synthetic retrievals to more effectively use compute time. Our posterior sample spread from the synthetic retrievals therefore includes any uncertainty introduced by calculating the covariances at a single ctP.
  - Changes made: p9L9—p9L11 now reads: ". We calculate covariances at a single value of P\_top, but the convergence of our synthetic retrieval tests across a range of true P\_top values shows that we obtain reliable results regardless."
- 10. Page 8, L10. what is the priori for cloud top pressure here? what is the error in OCO-2 measurement itself?
  - → Response: Clarification text added. We had provided information on the Ptop apriori but not the specific instrumental uncertainty relevant to these values. We have repeated the Ptop prior details so that hopefully readers don't miss them.
    - Changes made: p10L25—p11L2 now reads: "The squared OCO-2 radiance uncertainties are added to the diagonal elements of the observation error covariance matrix with no cross correlation. We use the standard OCO-2 version 7 uncertainties, and SNR increases as the radiance in a given channel increases. The median
- 40 SNR ranges from just over 400 for the  $\tau$  = 5 cases to around 700 for the  $\tau$  = 25 cases. The SNR reaches a minimum of 72 in an absorption band channel in a  $\tau$  = 5 case, and a maximum of 763 in a weakly absorbing channel in a  $\tau$  = 25 case.

Forty true cloud cases are used with five of each case where optical depth ranges from 5 to 40 in increments of 5 and cloud-top pressure is randomly selected to be between 680—900 hPa and rounded to the nearest 10 hPa.

- 45 The prior cloud properties are assumed to be unbiased, so are randomly sampled from a Gaussian with a mean equal to the truth and a standard deviation equal to the prior errors above. Each synthetic retrieval begins with a separate prior, and the prior is also used as the first guess."
  - 11. P10, L7. Do these 75 channels have the same wavelenghts for all test cases?
- 50

10

15

20

25

30

- Response: Yes, text added and we have added reminders and clarifications throughout. You're right that this is an important point!
- → Changes made: p5L6 text added: "but will select a consistent micro-window of the same channels for each."

P10L13—p10L15 now reads: "While this may result in a different location for each size of micro-window, the location is fixed for an individual case, i.e. the 5-channel microwindow consists of the same 5 channels in all 216 cases."

P12L18—p12L20 reads: "the 75 channel micro-window containing the OCO-2 channels 353—426 (indices counting from 1 for the full 1,016 OCO-2 L1bSc channels) consistently satisfies our P\_top and  $\Delta P_c$  criteria and reduces the full wavelength range from 759.2—771.8 nm to 763.5—764.6 nm." [inclusion of channel indices helps imply it's fixed]

New Figure 6 shows the selected channels.

12. P11, last sentence. what is proposed here is a strong statement. What is the basis to support that "assumptions made here don't affect primary conclusion" here?

15

10

5

- → Response: Fair point
- → Changes made: Text removed.
- 20 13. Finally, it is not all that clear if measurement in O2 A with such a finer spectral resolution will be needed? In other words, using 75 channels vs. using just one channel (such as from EPIC, MERES or TROPOMI) for cloud retrievals, are there huge differences? Answering this question will greatly improve the impact of this paper.
- Response: We now report the spectral resolution requirements given by O'Brien & Mitchell and Heidinger & Stephens. We also reproduced a calculation similar to Schuessler et al. and get the same answer: that GOME-2 spectral resolution isn't enough to obtain the three pieces of information we require. Our expanded introduction now explains that two A-band channels are insufficient (implicitly addressing MERIS, though we use POLDER as an example). We also now discuss TROPOMI it has the nominal spectral resolution, its larger spatial resolution will be a disadvantage in heterogeneous scenes however.

[revised manuscript text omitted]

In principle, obtaining the three properties  $\tau$ ,  $P_{top}$  and  $\Delta P_c$  might require only three independent channels. However, the optimal set would require knowing the cloud state beforehand, so we aim to select the smallest set of neighbouring channels.

5 that will allow accurate retrievals across the full range of cloud cases.. This approach aims to optimise a cloud property retrieval and due to limitations related to the radiative transfer implementation and computational burden, droplet size is not a retrieved property, instead it but contributes to the posterior uncertainty. It has been suggested that a Above-cloud CO2 retrievals have been found to require cloud can be retrieved if cloud properties are well known, including the effective droplet radiussize 
[revised manuscript text omitted]
  $S_T$ , humidity profile  $S_q$  and effective droplet radius  $S_{reff}$  such that:

$$\quad \mathbf{S}_{\boldsymbol{\epsilon}} = \mathbf{S}_{\boldsymbol{I}} + \mathbf{S}_{\boldsymbol{T}} + \mathbf{S}_{\boldsymbol{q}} + \mathbf{S}_{reff} \tag{10}$$

In reality, the temperature and humidity uncertainties are likely to be correlated, but this simplifies the calculation and allows unique attribution of covariance sources. The matrix  $S_I$  is a diagonal matrix so averaging over more channels reduces the total posterior uncertainty even if the Jacobians are not independent. Its elements are equal to the square of the instrumental uncertainty, which depends on the radiance.

- 10 For  $S_T$  and  $S_q$  we follow the approach of (Chang et al., 2017) and perturb the tropical, mid-latitude and high-latitude atmospheric profiles 2,000 times for temperature or humidity separately with: Uuncertainties are based on the 1 km resolution AIRS validation results (Divakarla et al., 2006). For temperature we add a uniform perturbation to each level with a value sampled from a zero mean ( $\mu$ ) Gaussian with standard deviation ( $\sigma$ ) of ±1.5 K. For specific humidity we sample from a zero mean Gaussian with a standard deviation of unity, then scale this value based on pressure level. The scaling is equivalent to
- 15 ±20 % of the initial specific humidity at the surface, increasing linearly to ±50 % of the layer values at 250 hPa and remaining at ±250 hPa% for levels with lower pressure. and are ±1.5 K in temperature and for specific humidity, scale linearly from ±20 % at the surface to ±50 % at 250 hPa. At higher altitudes, ±50 % is used at all levels. For each perturbed temperature simulation, the entire temperature profile is uniformly perturbed by a single value sampled from a zero mean Gaussian with standard deviation ±1.5 K. For specific humidity, we sample from a zero mean Gaussian with unit standard deviation and multiply that
- 20 value by the percentage scaling profile described above. The calculation was also performed with 2,000 perturbations applied to reff assuming by sampling from a lognormal distribution that approximates the effective radius distribution reported by MODIS for our November 2015 low cloud cases. This lognormal fit has an arithmetic mean of 12.0 μm, but after excluding values outside the 4—30 μm retrieved by MODIS, with a mean of This distribution has an arithmetic mean of 12.6 μm and 5—95 % range of the values fall within 7.5—19.4 μm the arithmetic mean is 12.6 μm and 5—95 % of the values fall within
- 25 7.5—19.4 μm. We choose reff = 12 μm in our default retrieval as we are restricted to integer values by the available L2RTM Mie scattering tables, and based on its similarity to the full distribution mean.
   For each set of perturbations, we simulated the A-band spectra for cloud optical depths of 5, 10 and 25 and solar zenith angles

of approximately 30°, 45° and 60° with a cloud-top pressure of 850 hPa. We calculate covariances at a single value of  $P_{top}$ . but the convergence of our synthetic retrieval tests across a range of true  $P_{top}$  values shows that we obtain reliable results

30 regardless.

The output was sampled with each of spectra are provided for each of the 8 different instrument line shapes associated with the 8 different OCO-2 across-track sounding positions.

For each set of 2,000 perturbed outputs, we estimated the covariance matrix elements,  $S_{i,j}$ , where i, j refer to channel indices, as:

$$S_{i,j} = \sum_{k} (I_{i,k} - \langle I_i \rangle) (I_{j,k} - \langle I_j \rangle) / N$$
(11)

Where the sum is over the *N*=2,000 spectra of radiance *I*, which are individually referred to using the index *k*. In this case <  $I_i > \text{and} < I_j > \text{are the sample mean radiances in the relevant pixels channels } i \text{ and } j$ .

**3.2 Channel selection**

Eq. (3) and Eq. (6) state that we can determine the information content and posterior error covariance from the prior covariance, observation covariance and Jacobians. Our aim is to select the optimal micro-window of consecutive OCO-2 pixels channels to provide a retrieval that efficiently reduces the posterior state error.

- 10 We use the L2FP radiative transfer model to simulate OCO-2 spectra for marine liquid clouds of  $\tau$  in [5, 10, 25] and  $\underline{P_{top}}$  in [680, 750, 850] hPa, for each of the 3 meteorological cases described in Sect. 3.1 and for each of the eight across-track sounding positions. In each case, the solar zenith angle is 45° and the Jacobians for  $\tau$ ,  $P_{top}$  and  $\Delta P$  are determined by finite differencing. The relevant observation covariance is that determined for the same sounding position, region and optical depth in Sect. 2.2 at SZA = 45°. Prior covariance is assumed to be diagonal, equivalent to an error of 1.5 in  $\tau$ -error of ±1.5, of 60 hPa in  $P_{top}$  of ±60

[revised manuscript text omitted]